# High-resolution species assignment of *Anopheles* mosquitoes using *k*-mer distances on targeted sequences

**Marilou Boddé[1,2]\*, Alex Makunin[2], Diego Ayala[3], Lemonde Bouafou[3], Abdoulaye Diabaté[4], Uwem Friday Ekpo[5], Mahamadi Kientega[4], Gilbert Le Goff[3], Boris K Makanga[6], Marc F Ngangue[7], Olaitan Olamide Omitola[5], Nil Rahola[3], Frederic Tripet[8], Richard Durbin[1,2], Mara KN Lawniczak[2]\***

[1]Department of Genetics, University of Cambridge, Cambridge, United Kingdom; [2]Wellcome Sanger Institute, Hinxton, United Kingdom; [3]Institut de Recherche pour le Développement, MIVEGEC, Univ. Montpellier, CNRS, IRD, Montpellier,, France; [4]Institut de Recherche en Sciences de la Santé, Direction Régionale de l'Ouest, Bobo-Dioulasso, Burkina Faso; [5]Federal University of Agriculture Abeokuta, Abeokuta, Nigeria; [6]Institut de Recherche en Ecologie Tropicale, Libreville, Gabon; [7]Centre International de Recherches Medicales de Franceville, Franceville, Gabon; [8]Centre for Applied Entomology and Parasitology, Keele University, Newcastle, United Kingdom

**\*For correspondence:**
mmb52@cam.ac.uk (MB);
mara@sanger.ac.uk (MKNL)

**Competing interest:** The authors declare that no competing interests exist.

**Abstract** The ANOSPP amplicon panel is a genus-wide targeted sequencing panel to facilitate large-scale monitoring of *Anopheles* species diversity. Combining information from the 62 nuclear amplicons present in the ANOSPP panel allows for a more senstive and specific species assignment than single gene (e.g. COI) barcoding, which is desirable in the light of permeable species boundaries. Here, we present NNoVAE, a method using Nearest Neighbours (NN) and Variational Autoencoders (VAE), which we apply to *k*-mers resulting from the ANOSPP amplicon sequences in order to hierarchically assign species identity. The NN step assigns a sample to a species-group by comparing the *k*-mers arising from each haplotype's amplicon sequence to a reference database. The VAE step is required to distinguish between closely related species, and also has sufficient resolution to reveal population structure within species. In tests on independent samples with over 80% amplicon coverage, NNoVAE correctly classifies to species level 98% of samples within the *An. gambiae* complex and 89% of samples outside the complex. We apply NNoVAE to over two thousand new samples from Burkina Faso and Gabon, identifying unexpected species in Gabon. NNoVAE presents an approach that may be of value to other targeted sequencing panels, and is a method that will be used to survey *Anopheles* species diversity and *Plasmodium* transmission patterns through space and time on a large scale, with plans to analyse half a million mosquitoes in the next five years.

## Editor's evaluation

This piece presents a new method, ANOSPP, that uses cheap sequencing to infer genealogical relationships between Anopheles individuals, including the possibility of species-level identification. The method is versatile and will be useful to vector biology researchers. The proof-of-principle presented in the manuscript is convincing and will serve as a blueprint for future studies in Anopheles.

## Introduction

The *Anopheles* genus contains dozens of mosquito species that are the vectors for the *Plasmodium* parasites which cause human malaria, and are thus of global public health interest. The genus contains nearly 500 formally described species (*Harbach and Kitching, 2016*), which span more than 100 million years of evolution (*Marinotti et al., 2013*). Only a subset of these species has the ability to transmit human malaria; however this vectorial capacity is not limited to a specific part of the species tree, but found throughout the genus phylogeny. Many species are members of closely related species complexes or groups, which are morphologically indistinguishable and share large amounts of genetic variation due to their ability to hybridise in areas of sympatry (*Anopheles gambiae 1000 Genomes Consortium, 2017*; *Harbach and Kitching, 2016*). Accurate species identification of *Anopheles* mosquitoes is difficult yet of crucial importance. This is because even closely related species may differ in vectorial capacity, insecticide resistance status and behavioural traits, all of which can influence the efficacy of malaria control efforts (*White et al., 2011*). Novel control efforts like gene drive are likely to be implemented this decade for *An. gambiae,* and the success of these efforts will need to be closely monitored. A thorough understanding of the geographic and temporal distribution of different *Anopheles* species and the potential extent of gene flow between species within the same complex is a necessary condition to implement gene drive technology as a malaria control tool.

Currently, the typical process of species identification for *Anopheles* mosquitoes starts by assigning them to species complexes or groups using morphological keys (*Gillies and Meillon, 1968*; *Gillies and Coetzee, 1987*; *Rattanarithikul and Panthusiri, 1994*; *Coetzee, 2020*; *Irish et al., 2020*). These morphological keys are usually specific to geographical regions and hence require in-depth and up-to-date knowledge of the species ranges. Moreover, the keys are specific to certain life-stages and may require one to grow larvae to a later stage or even adulthood. Because the morphological features distinguishing one group of species from another can be very nuanced, the accuracy of morphological classification depends on the level of experience and expertise of the person carrying out the identification. Species inside species complexes can be morphologically indistinguishable from each other at the adult stage and hence molecular assays are required for precise species identification. The most commonly used method is a PCR-based species diagnostic assay, targeting the highly variable internal transcribed spacer (ITS2) or similarly variable genomic regions (*Cohuet et al., 2003*; *Scott et al., 1993*; *Fanello et al., 2002*; *Wilkins et al., 2006*), although other approaches exist, for example based on mass spectrometry (*Nabet et al., 2021*). The PCR assays require the use of primers specific to the species complex or group, hence higher level morphological misclassification can lead to failure to generate PCR product or even erroneous species classification (*Erlank et al., 2018*). Mutations in primer or restriction sites can also lead to PCR failures. Moreover, in the case of hybrids or cryptic species, species identification based on a single marker can result in overconfident assignment to a single species, lacking the nuance that is desirable in this case. There exist panels targeting multiple loci; however, these are also specifically designed to work on a single species complex (*Rongnoparut et al., 1996*; *Lanzaro et al., 1995*; *Wang-Sattler et al., 2007*; *Santolamazza et al., 2008*).

To help overcome these challenges, a multilocus amplicon panel called ANOSPP (for '*ANO*pheles *SP*ecies and *P*lasmodium') was previously designed to amplify loci from any individual from any Anopheline species (*Makunin et al., 2022*). In brief, the panel targets 62 loci in the generic *Anopheles* nuclear genome, spread over all chromosome arms, including exonic, intronic as well as intergenic regions. Additionally, it targets two conserved loci on the generic *Plasmodium* mitochondrial genome, to simultaneously evaluate *Plasmodium* presence and species for each individual mosquito. Sequence data from up to 62 nuclear loci targeted by the ANOSPP panel for each sequenced specimen increases the resolution to distinguish closely related species, and to flag potential hybrid or contaminated samples as well as cryptic species. Moreover, the multilocus approach opens the possibility of population genetic and structure analyses for single or closely related species.

The ANOSPP panel has been developed to improve accuracy and depth of information as well as to drive down costs and time required to carry out vector species surveillance (*Makunin et al., 2022*). Accordingly, all that is required is to identify the individual mosquito as an Anopheline (as opposed to a Culicine), which requires minimal expertise as it is based on the length of the maxillary palps. Furthermore, the panel can use an extremely small aliquot of DNA (<1% of whole mosquito extraction) extracted from each mosquito using a cheap, nondestructive, high-throughput workflow (*Makunin et al., 2022*). Each mosquito is stored in the well of a 96well plate in ethanol, the

non-destructive lysis buffer is added, incubated overnight, and then removed. Ethanol is then added again to the mosquito carcass to preserve the mosquito and enable morphological evaluation of any individual post-sequencing. A dilution of the lysate is made, which is then PCR amplified with a cocktail of 64 primer pairs in a single well through a two-step process (*Makunin et al., 2022*). A single Illumina library is generated containing amplified and bar-coded material pooled from 768 samples, then sequenced on a single Illumina MiSeq lane.

The ANOSPP panel originally used a species assignment method based on alignment distances, using static sequence similarity thresholds fitted per target (*Makunin et al., 2022*). Using this method, amplicons contain sufficient information to distinguish between the majority of the 56 species represented in the original dataset; only samples within some species complexes could not be accurately assigned. However, this method was only tested on a small dataset and the test set was also used as the training set to form the haplotype clusters. Additionally, some targets were extremely divergent between different groups of samples in the dataset and the distances for these targets appear to be sensitive to the choice of alignment algorithm. Here, we present a new species assignment method that we call NNoVAE, which is a $k$-mer-based method consisting of an initial step using Nearest Neighbours (NN) that identifies samples down to a species or, in some cases, a species complex, and a second step using a Variational Autoencoder (VAE) for species identification within a species complex. The VAE in the second step is specifically trained on the species complex to which test samples get assigned by the NN step. The complex-specific VAE is required because the similarity between closely related species is too close to detect with a general purpose method like the NN step. Both assignment steps in NNoVAE work on the same $k$-mer tables.

NCBI GenBank's BLAST (*Altschul et al., 1997*) can be used to assess the similarity of any sequence to those within any INSDC database such as NCBI GenBank (*Benson et al., 2018*). BLAST works on a single query at a time, and does not assign species directly, but rather reports the best hits of the query sequence alongside p-values based on the alignment score corrected for the size of the database. Similarly, the standard search methods for BOLD (*Ratnasingham and Hebert, 2007*) performs species assignment based on a threshold of percent identity between the test sample and the samples in the reference database. The threshold is informed by extensive knowledge of the divergence of the single marker cytochrome oxidase subunit I (COI) between different species. In addition, the BOLD analysis toolkit contains a NN method to explore the relationship between multiple species in the database. Like BLAST and the BOLD toolkit, the NN step of NNoVAE uses sequence-based distances between the haplotypes of test samples and samples in a reference database; however, this is performed across all amplicons and uses $k$-mer distances instead of alignments. By using $k$-mers, we avoid the issues associated with alignment of highly divergent sequences and moreover are able in a natural way to use short indels and small structural variants as well as SNPs. NNoVAE aims to assign species identity and simultaneously collect information about the relationship of the test sample to different species or species-groups in the database, which is particularly useful when the test sample is of a species not represented in the database. Moreover, NNoVAE combines the information from the 62 different target regions in ANOSPP to build confidence in the species assignments, or reflect uncertainty in the species assignments where appropriate.

NNoVAE resolves species identity within the *An. gambiae* complex using a variational autoencoder trained on samples from this complex. A VAE is a machine learning method that learns structure in high-dimensional data by encoding it into a low-dimensional space and subsequently generating simulated data from the low-dimensional encodings (*Kingma and Welling, 2013*). VAEs have previously been used for species delineation in spiders (*Derkarabetian et al., 2019*) and visualisation of population structure in *Anopheles* and humans (*Battey et al., 2021*). Both these studies used sequence alignments containing much more genomic sequence than the amplicon panel provides. In contrast, NNoVAE is $k$-mer based, making it robust to alignment ambiguity and enabling it to efficiently make use of all the available sequencing data.

With the application of the NNoVAE method to data resulting from the ANOSPP panel, we aim to create a robust and efficient platform for molecular species identification within the entire *Anopheles* genus. We present a method that can assign individuals to any of the 62 species currently represented in the reference database as well as taxonomically place species not yet represented. We will include more samples from more species in the reference database as they are sequenced by the amplicon panel, so as to represent the full diversity of the *Anopheles* genus. NNoVAE also indicates

the uncertainty of the assignment and provides information on other closely related species in the dataset, so in particular can flag potential hybrids. It can also be applied to whole genome shotgun (WGS) data by computationally extracting the amplicon target regions, allowing to integrate WGS reference panels with amplicon-sequenced field samples.

## Results

### Reference database construction

Species assignment methods commonly work by comparing a query sequence to a reference database (*Ratnasingham and Hebert, 2007*; *Benson et al., 2018*). The completeness and quality of the reference database heavily influence the accuracy of the assignment method. The reference database we constructed consists of well-curated samples sequenced with the panel, in silico extracted reference genomes, and in silico extracted whole genome short read data. As we expect that the reference database will expand to include additional species and populations over time, we assign a version number to the database. The reference database described here, which we call NNv1, contains 186 samples, representing 62 species spread over 4 subgenera. The dataset from *Makunin et al., 2022* forms the backbone of the reference panel. In addition, six species in the *An. gambiae* complex have been included from publicly available whole genome data (*The Anopheles gambiae 1000 Genomes Consortium, 2021*; *Fontaine et al., 2015*) in order to increase the resolution in a group of hard to distinguish species. To maintain the advantages of multi-locus assignment, we required samples to have at least 10 targets amplified to be included in the reference database. Ideally, the database would contain several specimens per species to represent within species variation. This is particularly important for species with a wide geographical range.

The amplicon panel is designed to improve accuracy of species assignments over morphological or single-marker methods. However, the species labels supplied by our sample partners were in most cases obtained by the latter methods, hence it was necessary to reconfirm them. All label information for the reference database NNv1 is listed in *Supplementary file 1*, and the assignment of species labels is discussed in more detail in Appendix 1, but we present an outline of the principles we used here. For most samples sequenced by the amplicon panel, two molecular barcodes (COI and ITS2) were also sequenced and compared to the sequences available in BOLD (*Ratnasingham and Hebert, 2007*) and NCBI (*Benson et al., 2018*). Here, we used the barcode information as well as the pairwise *k*-mer distances between samples in the reference database NNv1 to generate a consensus species label for each sample. A few samples show inconsistencies between their partner labels, molecular barcodes and amplicon assignments; these samples were also removed or flagged as overly diverged in *Makunin et al., 2022*. For these samples, the distances to samples of the same species label is much larger than the distance to some other samples in the database, suggesting they were mislabelled. In some cases, there is additional evidence from molecular barcodes and it was possible to relabel them to the species they match. Other samples are clearly different from samples with the same species label, but do not clearly belong to any other species in the database. These are labelled as the species-group they belong to appended by '_sp1', '_sp2', etc. Hopefully, by extending the reference database in the future, we can get a better understanding of which species such samples represent. For some closely related species, the different species labels were supported neither by the pairwise-distances nor by the barcodes. Samples from these species are assigned to a group of closely related species and their consensus labels end in '_c' to notify that they are part of a complex of species that cannot be distinguished by the nearest neighbour method. None of the in silico extracted samples showed inconsistencies with pair-wise *k*-mer distances and hence all retained their published labels.

### Species-groups construction

The species-groups are defined based on the pairwise *k*-mer distances between samples in the reference database NNv1 and make use of the consensus species labels discussed above. The *k*-mer distance between two samples, $s_1$ and $s_2$, is defined as $d_{s,k}(s_1, s_2) = \frac{1}{|T_1 \cap T_2|} \sum_{t \in T_1 \cap T_2} \sum_{q_1 \in Q_{1,t}} \sum_{q_2 \in Q_{2,t}} \frac{d_k(q_1,q_2)}{|Q_{1,t}||Q_{2,t}|}$, where $T_i$ is the set of targets amplified in sample $i$ and $Q_{i,t}$ is the set of unique haplotypes of sample $i$ at target $t$ and by $|S|$ we mean the number of elements in set $S$. The *k*-mer distance $d_k$ between two sequences is defined in the Methods section on *k*-mers. So in words, for a given target the *k*-mer

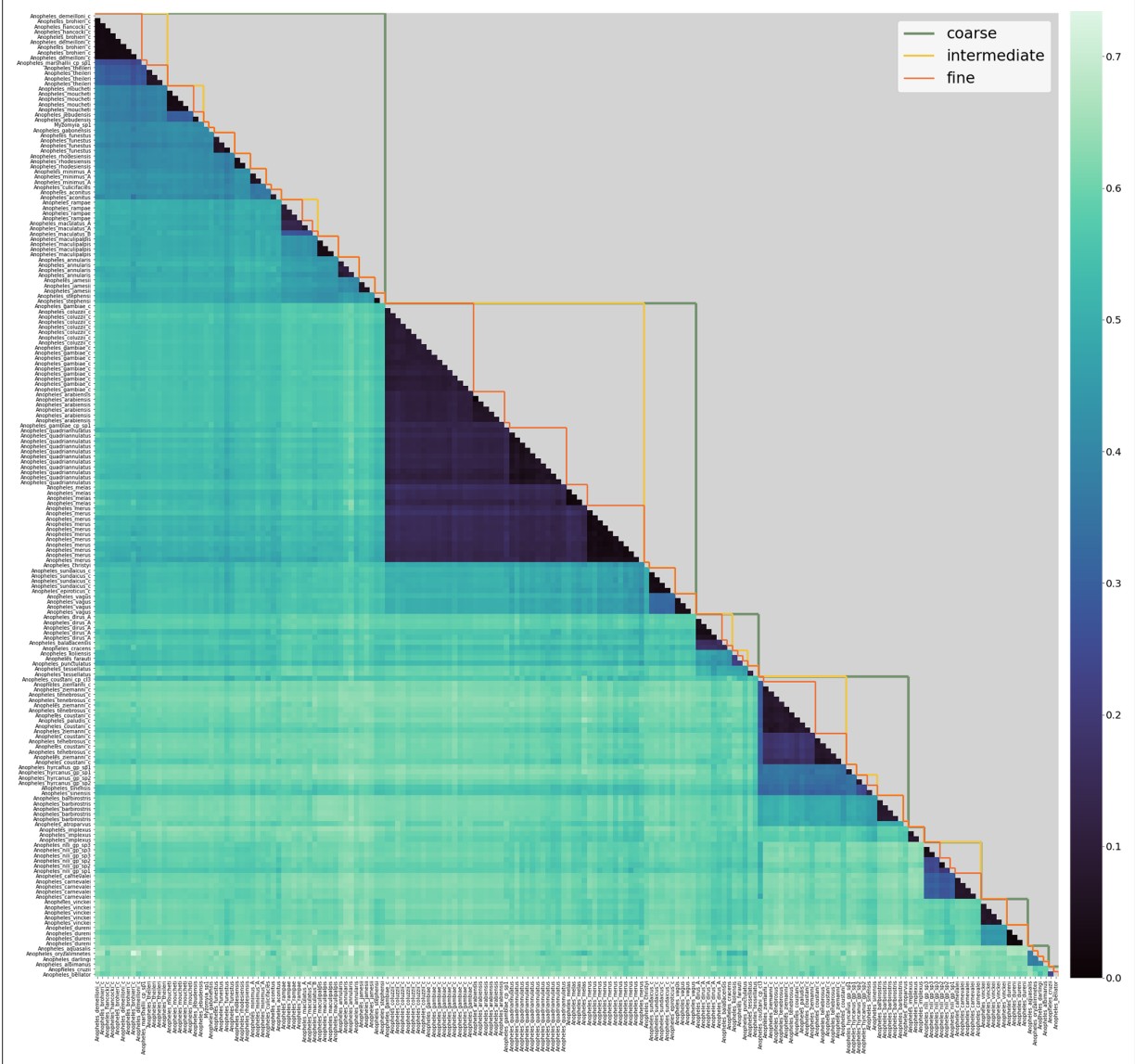

**Figure 1.** Lower triangle: heatmap of 8-mer distances between pairs of samples in the reference database. Samples are on the x- and y-axis, roughly ordered by phylogeny and labelled with their consensus species label. Dark colours correspond to small 8-mer distances and light colours to larger 8-mer distances. Upper triangle: species-groups at fine, intermediate and coarse levels (see main text for definitions of these).

The online version of this article includes the following figure supplement(s) for figure 1:

**Figure supplement 1.** Thresholds used to define species-groups.

distance between two samples is the mean $k$-mer distance between all pairs of haplotypes from the two samples; for example if sample $s_1$ is homozygous and $s_2$ heterozygous at target $t$, then the $k$-mer distance at target $t$ between these samples is the mean of the $k$-mer distance of the haplotype from $s_1$ compared to the first haplotype from $s_2$ and the $k$-mer distance of the haplotype from $s_1$ compared to the second haplotype from $s_2$. If a sample has more than two alleles at a single target, we take into account all haplotypes according to the above definition. The $k$-mer distance between two samples is defined as the average of the $k$-mer distance between these samples at all the targets that were amplified in both samples.

*Figure 1* shows the pairwise $k$-mer distances between all samples in the reference database. The samples are ordered roughly by phylogeny (as in the tree in *Figure 2*) and this results in a visible structure in the distance plot. One can observe dark triangles below the diagonal, reflecting that

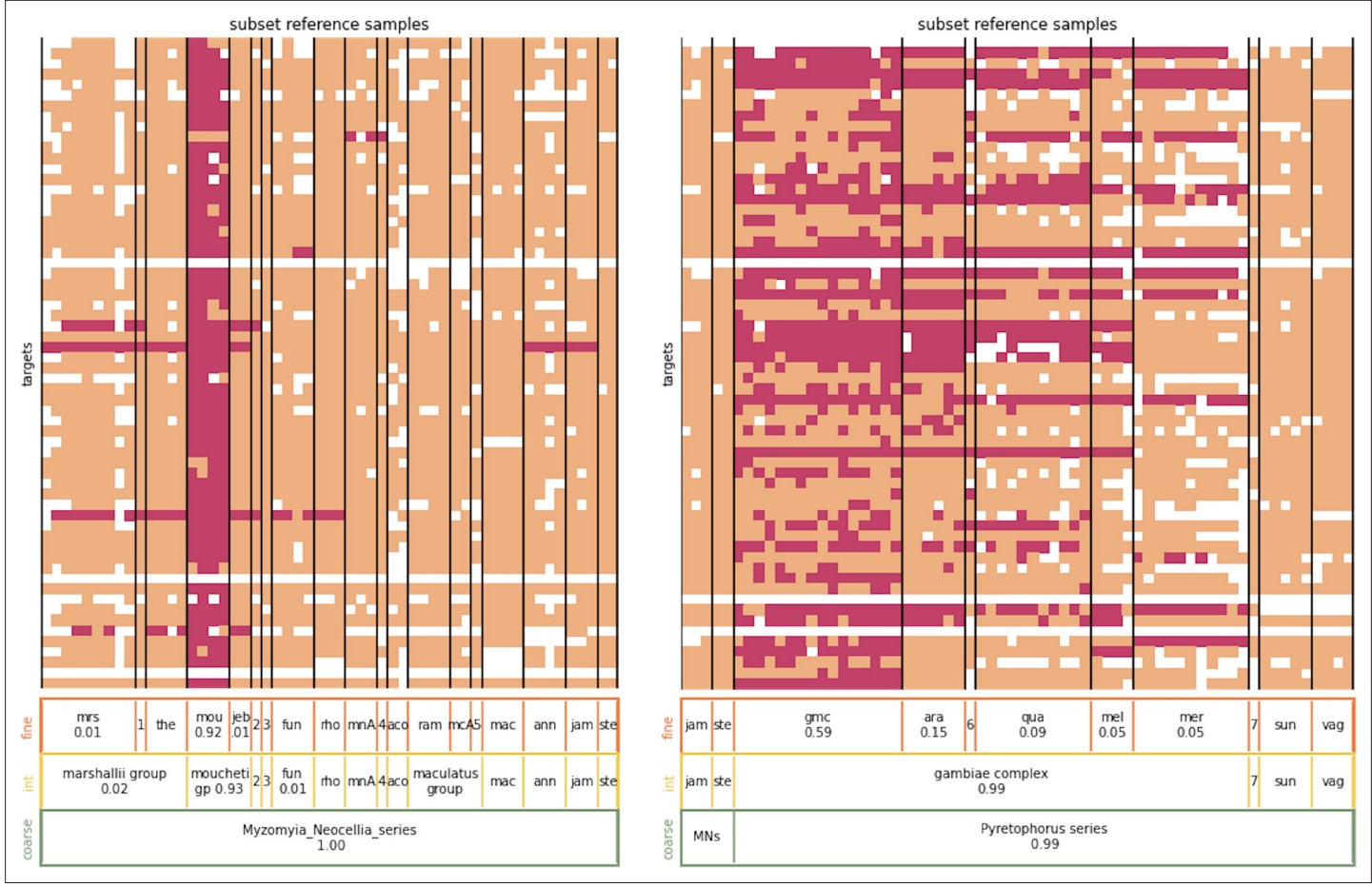

**Figure 2.** *Nearest neighbour assignment example.* **Left panel:** The heatmap shows the nearest neighbours of sample Amou-2–3, an *An. moucheti* specimen, at its different targets. For clarity, not all samples in the reference database have been displayed, only those in the *Myzomyia* and *Neocellia* series (except Amou-2–3). The samples from the reference database are arranged along the x-axis and the targets along the y-axis. An entry is coloured pink if the corresponding sample from the reference database has a nearest neighbour sequence at the corresponding target. Peach entries indicate that the corresponding sample from the reference database does not carry a nearest neighbour sequence at the corresponding target. If either the test sample or the reference sample did not amplify at the corresponding target, the entry is white. The bars at the bottom show the overall assignment proportions for the displayed species-groups, only assignment proportions of at least 1% are shown. From top to bottom the assignment proportions are for the fine, intermediate,, and coarse level. For the three-letter code abbreviations of species-groups, see ***Supplementary file 1***. The numeric abbreviations stand for 1: *An. marshallii complex sp1*, 2: *Myzomyia sp1*, 3: *An. gabonensis*, 4: *An. culicifacies*, 5: *An. maculatus B*; none of these species-groups represent more than 1% of the assignment. **Right panel:** The heatmap showing the nearest neighbours of sample Agam-35, an *An. gambiae* specimen. Not all samples in the reference database are displayed, only those in the *Pyretophorus* series (except Agam-35) as well as five samples from the *Neocellia* series. The numeric abbreviations stand for 6: *An. gambiae complex sp1* (0.06 assignment proportion), 7: *An. christyi* and MNs stands for *Myzomyia_Neocellia_series*.

samples of the same or closely related species have a smaller *8*-mer distance to each other than to other samples. This effect repeats itself on different scales, mirroring the multi-level structure in the phylogenetic tree.

We identify clusters of samples in the reference database NNv1 and we will refer to those as species-groups. It is important to emphasise that the way we use the term 'species-groups' does not refer to a taxonomic classification, but to the clusters of samples defined by thresholds on the *8*-mer distances between samples. Ideally, a threshold would partition the samples in the reference database NNv1 into species-groups, such that the *8*-mer distance between each pair of samples within the group is smaller than the threshold and the *8*-mer distance from any sample within the group to any sample outside the group is larger than the threshold. The motivating assumption is that the samples in the same species-group share more recent ancestry with each other than with samples that are members of different species-groups. While several thresholds satisfy the partitioning condition,

they are most useful for the purpose of species classification when they generate species-groups that (approximately) correspond to classification at an established taxonomic level. Not surprisingly, there are no thresholds that perfectly satisfy the partitioning condition and are completely concordant with current taxonomy, but several thresholds give groups that are close to taxonomic entities and given the paucity of molecular data for some of these species, it may be that the taxonomic entities are less phylogenetically accurate than the *k*-mer based groupings. In exploring different thresholds (*Figure 1—figure supplement 1*), we selected three levels that best matched described taxonomic entities. The species-groups at each of the threshold levels we have selected are listed in *Supplementary file 1*.

For very low thresholds, each sample will form its own species-group, which is not informative. For a threshold of 0.1 on the sample *8*-mer distance, most species-groups satisfy or nearly satisfy the partitioning condition, and each species-group contains a single species or multiple species from a known species complex or group, for example *An. gambiae* and *An. coluzzii* are grouped together at the 0.1 threshold. We refer to the species-groups at the 0.1 level as the fine level species-groups and they are most useful for species assignment, because they provide the highest resolution. A threshold of 0.3 merges samples representing species from some well-known complexes, like the entire *An. gambiae* complex, into a single species-group. Similarly, many of the species for which we currently only have a single representative in the reference database NNv1, are merged into larger species-groups. We refer to this level of species-groups as the intermediate level, which provides insight into the degree of similarity between different fine level species-groups. Additionally, the intermediate level species-groups can be informative when we sequence a sample whose species is not represented in the database because the species assignment at the intermediate level places the sample within its most closely related species-group in the database. Thresholds higher than 0.3 tend to violate the partitioning condition to a greater extent, but it is desirable to include a coarse level classification to get an approximate taxonomic position of unrepresented species and of more diverged species. We selected 0.51 as a threshold for the coarse level classification because this gives reasonably clear species-groups that roughly correspond to taxonomic series. However, it is not perfect, as it groups together the *Myzomyia* and *Neocellia* series in the *Cellia* subgenus and it splits the *Neomyzomyia* series into three distinct species-groups. A similar effect was observed in *Makunin et al., 2022*, where the *Neomyzomyia* series did not form a monophyletic clade.

At the fine level, most species-groups contain all samples from a single species, but there are some exceptions. For some species, for example *An. nili* and *An. hyrcanus*, the samples are split into multiple fine-level species-groups, because they appear much more distinct from each other than you would expect in a single species (see Appendix 1 and *Figure 1—figure supplement 1* for more detailed discussion). In our assignment, these will be treated as distinct species, highlighting to entomologists and taxonomists that further work to refine species in these groups is needed. Conversely, there are also species-groups that contain samples from multiple different species. These do not represent a single species, but instead they represent a complex of closely related species. Species inside species complexes often share a lot of genetic variation and the *k*-mer distance based method that we discuss here does not have sufficient resolution to reliably distinguish between them. The species-groups at the fine level that contain more than a single species are the *An. marshallii* group (contains *An. hancocki, An. brohieri,* and *An. demeilloni*), *An. gambiae/coluzzii*, the *An. sundaicus* complex (contains *An. sundaicus* and *An. epiroticus*) and the *An. coustani* group (contains *An. coustani, An. ziemanni, An. tenebrosus,* and *An. paludis*).

## Nearest neighbour assignment

The first step of the hierarchical assignment method performs nearest neighbour assignments to samples in the reference database at the three different levels of species-groups introduced above. The assignment is initially done independently at each target, for each sample, computing assignment proportions for the species-groups at the chosen level, normalised such that they sum up to one over all species-groups. The resulting per-target assignment proportions are then averaged over all targets, resulting in the overall sample assignment proportions at the chosen level. If the sample assignment proportion is at least 0.8 for one species-group, the sample is classified as a member of that group. If the classification threshold of 0.8 is not met at the chosen level, the sample remains unassigned at that level.

For example, to assign a sample $s$ at the coarse level, we translate its target sequence corresponding to target 1 to an $8$-mer count table, denoted as $q_{s,1}$. Next, we compute its 8-mer distance, $d_8\left(q_{s,1}, q_{r,1}\right)$, to every target sequence in the reference database corresponding to target 1, that is to every $q_{r,1}$. The nearest neighbours of the test sequence $q_{s,1}$ are those sequences in the database that minimise the 8-mer distance between themselves and $q_{s,1}$. In other words, the nearest neighbour sequences of $q_{s,1}$ are the target 1 sequences in the database with the highest percentage of matching $8$-mers. The nearest neighbour sequence of $q_{s,1}$ can be a sequence that occurs in a single sample in the reference database, or the nearest neighbour sequences can be the same sequence occurring in multiple samples in the database, or the nearest neighbour sequences can be distinct sequences that have the same distance to $q_{s,1}$.

Now we bring in the species-groups. For each species-group, we record the frequency of nearest neighbour sequences. This can be thought of as an 'allele-frequency' when we classify each target sequence as either a 'nearest neighbour allele' or not a 'nearest neighbour allele'. But just like allele frequency, it takes into account the zygosity of the samples. The nearest neighbour frequencies are normalised, such that they are equal to one when summed over all species-groups. These quantities are the per-target assignment proportions.

This procedure is repeated for every amplified target in the test sample. Finally the per-target assignment proportions are averaged over all successfully amplified targets to give the overall assignment proportions for sample $s$ at the coarse level. If there is a species-group with an assignment proportion of at least 0.8, the sample is classified as a member of this group, otherwise it remains unassigned at this level. Assignments to the intermediate and fine levels are made in the same fashion, starting from the same nearest neighbour assignments, but based on the relevant species-group memberships.

The per-target assignment proportions are based on the frequency of nearest neighbour sequences in the species-groups and not simply on the count of nearest neighbour sequences. The use of frequencies corrects for the different sizes of the species-groups. Suppose a nearest neighbour sequence occurs in 2 out of 10 samples of species-group A and in 1 out of 2 samples in species-group B (and assume all samples are homozygous). By using counts, we would attribute 2/ (2+1)=0.67 and 1/ (2+1)=0.33 assignment proportion to species-group A and B, respectively. But if we did this, species-group A would only have a higher assignment proportion because it contains more samples. What we are really interested in, is how similar the target sequence of the test sample is to the target sequences in the species-groups. So by using the nearest neighbour frequencies, the assignment proportions are (2/10)/(2/10+1/2)=0.29 and (1/2)/(2/10+1/2)=0.71 for species-group A and B, respectively.

The nearest neighbour frequencies observed for a given target of a given sample at a given assignment level are normalised to obtain the per-target assignment proportions. This normalisation ensures that every target is weighted equally. Without the normalisation, the weight of a target would be determined by a combination of the overall frequency of the nearest neighbour sequences and their distribution amongst the species-groups, whilst we are more interested in the distribution than the total frequency.

Targets that did not amplify in the sample are simply ignored. There are different reasons why a certain target does not get amplified in a sample. It might be that the primer binding sites for the target are too diverged or altogether absent in the test sample's genome. Or the target might not be amplified due to technical reasons like poor DNA quality. In the former case, we are implicitly using the information contained in the missingness, because we restrict our attention to the targets that did amplify in the sample, which should be the same targets that amplified in the samples of the same species contained in the reference database. In the latter case, as long as the missingness is randomly affecting the targets, ignoring missing targets does not bias the assignment proportions.

So far, we have assumed that the test sample was homozygous at each target. To generalise to the heterozygous case, we compute the per-target assignment proportions separately for both target sequences and average them for the final per-target assignment proportions. It does occasionally happen that a sample has more than two different target sequences. This can be due to errors in the PCR amplification or sequencing, contamination by other samples, or a certain target region might be duplicated in the genome of some species. In the NNv1 reference database on average 1.2% of targets per sample have more than two different sequences; in the query datasets discussed later in this article this percentage ranged from 0.3% to 2.3%. We deemed this small enough to simply extend

the per-target assignment proportion computation to include targets with more than two different sequences, by taking the average assignment proportions over different sequences as for the heterozygous case.

*Figure 2* shows two examples of the nearest neighbour assignment of a test sample. The test samples are Amou-3–2 and Agam-35, an *An. moucheti* and *An. gambiae* individual, respectively. For Amou-3–2 we see that for most targets, the nearest neighbour sequence is found in all four *An. moucheti* samples in the reference database NNv1. For some targets, the nearest neighbour sequence is only carried by a subset of the *An. moucheti* samples in the database, whilst for other targets the nearest neighbour sequence is also carried by individuals of other species. For Agam-35 the heatmap shows that, for many targets, the nearest neighbour sequences are not only found in *An_gambiae_coluzzii* samples, but also in samples from other species in the *An. gambiae* complex. There are only two matches to samples outside the *Pyretophorus* series, not shown here. This results in a high-confidence assignment to the *Pyretophorus* series at the coarse level as well as a high-confidence assignment to the *An. gambiae* complex at the intermediate level. At the fine level, the largest assignment proportion is to the *An_gambiae_coluzzii* species-group, but it does not meet the 0.8 classification threshold because of the relatively high assignment proportions to other species-groups within the *An. gambiae* complex. So at the fine level, this sample cannot be classified with sufficient confidence to a single species. Later, we will present a method to resolve the species identities of samples within the *An. gambiae* complex.

The species-group assignment has been tested on the reference database itself, by dropping out one sample at a time. The majority of samples could be assigned to the correct species-group at the fine level when using a threshold of 0.8 assignment proportion, see *Figure 3*, *Figure 3—figure supplement 1* and *Supplementary files 2–4*. To provide context, we have included a phylogenetic tree constructed from pairwise *8*-mer distances using FastME (*Lefort et al., 2015*) and displayed using TreeViewer (*Bianchini, 2021*). If we ignore the samples that form a species-group on their own, because we do not yet have sufficient representation for those species, 61.8% of samples are assigned correctly at the fine level, and 98.8% and 100% at the intermediate and coarse level respectively. In most cases, the fine level species-groups consist of a single species, although in some cases they comprise multiple species. The jump in assignment success from the fine to the intermediate level is mostly caused by the *An. gambiae* complex, which is well-represented in the reference database. Most samples within the complex can be assigned to the correct fine level species-group to some extent, but they only meet the assignment threshold at the intermediate level species-group assignment, where all samples in the *An. gambiae* complex are grouped together. This effect is seen in a few other groups as well, and motivates the VAE part of our assignment procedure (described below).

For the same group of samples, the average correct assignment proportion per sample at the coarse level is 99.4%. This shows that for samples of species that are well-represented in the reference database, there is a near perfect assignment at the coarse level. And in fact, the average correct assignment proportion of 95.8% at the intermediate level shows that these assignments are also generally with high confidence. At the fine level the confidence starts to break down for some samples, in particular the *An. gambiae* complex, which is represented by many samples in the reference database. Then the average correct assignment per sample is 81.3%. This shows that an additional classification method for species complexes is desirable.

In NNv1, 21 species-groups at the fine level consist of a single sample. These cannot be assigned to the correct species-group in the drop-out assignment experiment. 12 of these samples are members of a larger species-group at the intermediate level and 11 of them can be assigned at the intermediate level, the other sample is not assigned at the intermediate, nor at the coarse level. The remaining nine samples only become a member of a larger species group at the coarse level. Six of them can be classified at this level, the other three remain unclassified.

The four unclassified samples are *An. christyi*, *An. atroparvus*, *An. oryzalimnetes,* and *An. cruzii*. All of these are quite diverged from everything else in the reference database and as such do not exhibit a strong matching to any of the coarse level species-groups. In particular, the *Kerteszia* and *Nyssorhynchus* subgenus are underrepresented, both in number of species and number of samples, but also the basal species in for instance the *Pyretophorus* series are not well represented and are too diverged from the other species in this series to exhibit strong similarity to the other samples from this series. In the case of *An. cruzii* in the *Kerteszia* subgenus it is actually impossible to assign it to its coarse level

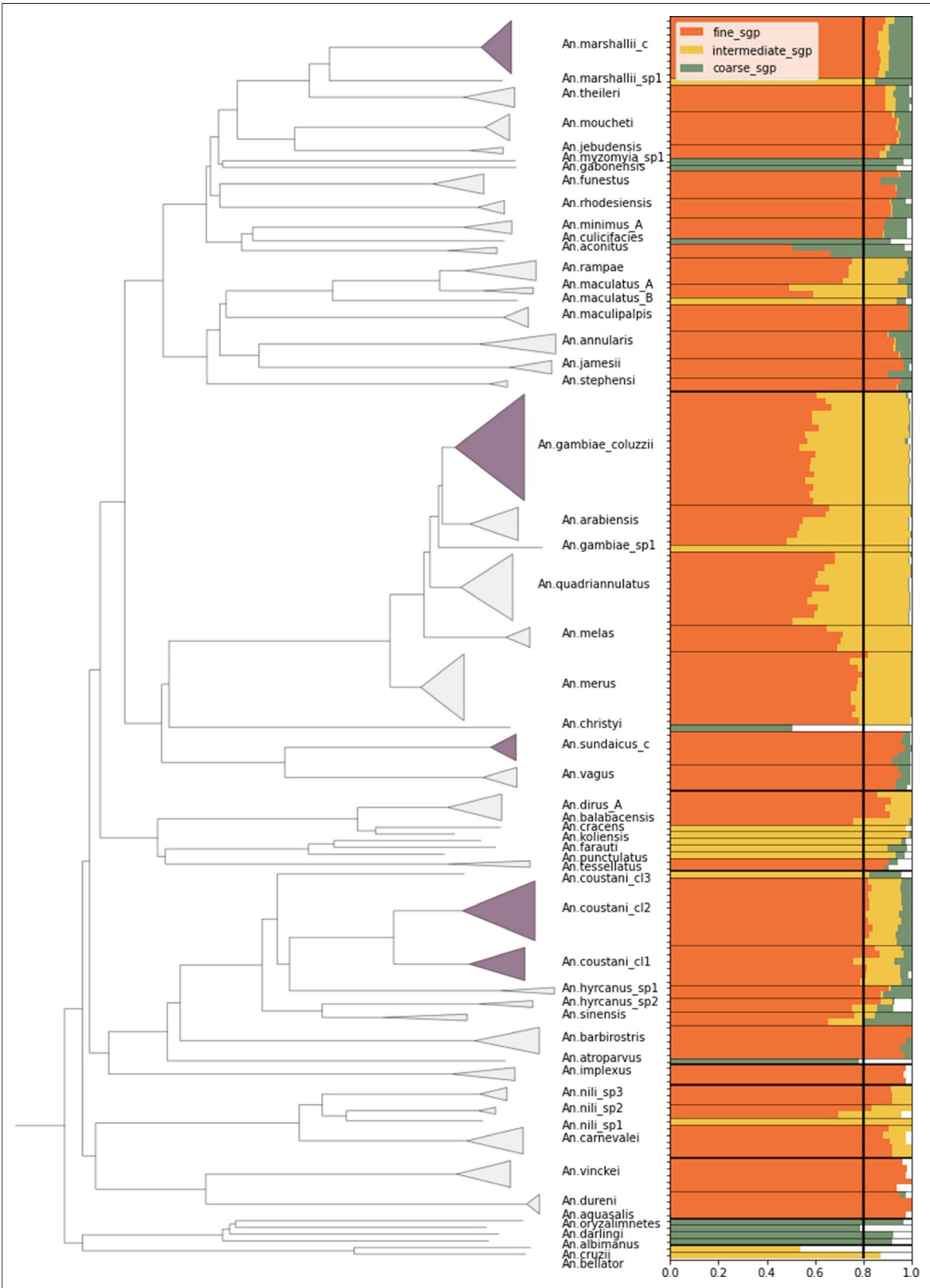

**Figure 3.** Species-group assignment accuracy on reference database NNv1. Samples were dropped out of the database one at a time to test the assignment accuracy. Left: phylogenetic tree of the samples in the reference database NNv1 constructed from pairwise *8*-mer distances using fastme. Samples are labelled by their fine level species-group label. Dark-shaded clades are instances of species-groups that contain more than one species. **Right:** Samples are placed along the vertical axis, ordered by the species tree. The bars represent the assignment proportion to the correct species-

*Figure 3 continued on next page*

*Figure 3 continued*

group and the colours indicate the species-group level. As an example, the first sample is assigned to the correct species-group with a proportion of 0.88 at the fine level, with a proportion of 0.91 at the intermediate level and with a proportion of 0.99 at the coarse level. The thin horizontal lines indicate the different species-groups at the fine level and the thick horizontal lines at the coarse level. The separation of the species-groups at the intermediate level has not been displayed for clarity. The vertical line represents the assignment threshold of 0.8.

The online version of this article includes the following figure supplement(s) for figure 3:

**Figure supplement 1.** Species-group assignment proportions.

species group with the current threshold of 0.8, because of its 26 amplified targets, only 15 are also amplified in the single other sample in the reference database from the *Kerteszia* subgenus, which results in a theoretical maximum assignment to the correct species-group of 0.58. Note that this is the only species-group at the coarse level for which the theoretical maximum correct assignment for a sample in the reference database is smaller than the threshold. But it again underlines the necessity to extend the reference database, both in number of species and number of samples per species.

When a sample is of a species not represented in the reference database, three things can happen. If its species is much more closely related to a single species in the database than to all the others, it will likely be assigned to the species it is related to. Alternatively, if the database contains multiple closely related species, it will be assigned at a higher level to the group that contains all these closely related species. If the sample is highly diverged from all species represented in the database, its nearest neighbours will essentially be chosen at random, and the assignment threshold will not be met. This emphasises the importance of extending the reference database, both by increasing the number of species represented and by increasing the number of samples per species, with a particular focus on capturing the within-species diversity (e.g. representing the geographic species range, representing all karyotypes when polymorphic chromosomal inversions are present).

The species-groups with a single representative can be used to explore the three possible scenarios for when the test sample belongs to a species not present in the reference database: assignment to a closely related species, assignment to a group of species at the intermediate or coarse level, or no assignment.

The first scenario is represented by *An.bellator*. The only other representative of the *Kerteszia* subgenus in our reference dataset is a sample representing *An. cruzii*. At the fine level species-groups, the *An. bellator* sample is assigned for 0.875 to *An. cruzii* and hence meets the threshold to be classified at the fine level. So it can happen that a sample is classified to the wrong species, if this species is the only reasonably close species in the database. This scenario is more likely for diverged groups of species with little representation in the database, for example the *Nyssorhynchus* and *Kerteszia* subgenera.

An example of the second scenario is the sample Amar-3–1, in the *Myzomyia* series. At the fine level, it has an assignment proportion of 0.474 to the *An_marshallii_complex* and 0.378 to the *An. theileri* species-group. All other species-groups have an assignment proportion of less than 0.06. So this sample is more related to these two species-groups than to anything else in the reference database, but it does not belong to either of them. At the intermediate level, it has an assignment proportion of 0.850 to the *An_marshallii_group* species-group, which is the *An_marshallii_complex* and *An. theileri* species-group combined. Now it meets the threshold and it will be classified as a member of the *An_marshallii_group* species-group.

*An. christyi* is an example of a sufficiently diverged sample that does not reach high assignment proportions for a single species-group at any level. At the coarse level, it is assigned to the *Pyretophorus_series* with 0.506 and the *Myzomyia_Neocellia_series* with 0.392 and other species-groups have much lower assignment proportions. Hence, it is not possible to classify this sample as a member of any species-group. Adding more samples from this and other underrepresented species to the database, would increase its power to classify samples from those species.

## Gambiae complex classifier datasets

The nearest neighbour approach is not able to confidently distinguish between closely related species that share a lot of genetic variation, but it does identify samples of those species as members of a species complex at the intermediate assignment level. To resolve the species identity inside these species complexes, we use a variational autoencoder approach specifically trained for the complex

under consideration. We demonstrate this method for the *An. gambiae* complex, both because many of our samples fall within this complex and it is medically relevant to be able to distinguish them, and also more practically, because we have access to a large dataset of species-labelled samples. The classifier we present here contains 7 out of 8 formally named species in the complex (*Coetzee et al., 2013*), as well as two putative cryptic species (*Tennessen et al., 2021*; *Barrón et al., 2019*). Compared to NNv1, three additional species are represented in this classifier to present as much of the diversity of the *An. gambiae* complex as possible. We expect that this method can be applied to other species complexes for which large species-labelled datasets are available.

The *An. gambiae* complex classifier has been constructed using a training set (GCref v1) and a validation set (GCval v1) of species-labelled samples. We included as many described species in the *An. gambiae* complex as we could find or generate sequence data for. For species with wide geographic ranges and a large amount of genomic data available, we also aimed to represent the diverse geography where possible.

Both GCref v1 and GCval v1 consist of amplicon sequences and in silico extracted published samples (*Fontaine et al., 2015*; *Neafsey et al., 2015*; *Tennessen et al., 2021*; *The Anopheles gambiae 1000 Genomes Consortium, 2021*). The samples from Nigeria and Madagascar were not species-labelled, but they were unambiguously classified by an earlier version of this classifier and we included these samples because Nigeria and Madagascar fill geographic gaps in our sampling dataset. The species represented in GCref v1 are *An. gambiae (406), An. coluzzii (222), An. arabiensis (94), An. quadriannulatus (11), An. melas (3), An.merus (6), An. bwambae (3), An. tengrela (38)*, and *An. fontenillei (4)*. These samples are generally high coverage: 97% of samples have at least 55 of 62 targets amplified. The average number of targets tends to be lower for the samples representing species other than *An. gambiae, An. coluzzii*, or *An. arabiensis*, which are also those species represented by fewer samples, but the geographic ranges of these other species are also much more restricted so the samples we do have are likely good representatives of the species. The species represented in GCval v1 are *An. gambiae (80), An. coluzzii (15), An. arabiensis (30), An. melas (1), An. merus (5)*, and *An. tengrela (12)*. The average number of amplicons for the species other than *An. gambiae, An. coluzzii* or *An. arabiensis* is lower than in GCref v1 set. Given that for those species, there is only a small number of samples available, we decided to use ones with at least 45 targets amplified in GCref v1 and the ones with at least 30 targets in GCval v1. Sample information for these datasets can be found in *Supplementary files 5 and 6*.

The input for the VAE is one *8*-mer count table per sample, summed over all targets. If a test sample is heterozygous at a given target, we translate each of its haplotypes to an *8*-mer count table and sum them to get the test sample's *8*-mer count table for the corresponding target. If a test sample is homozygous at a given target, we translate its haplotype to an *8*-mer count table and double the counts to obtain its *8*-mer count table at the corresponding target. The counts are doubled in order to represent the target sequences as diploid sequences, and not introduce artificial differences in the total number of *8*-mers between homozygous and heterozygous target sites. If the test sample has more than two different haplotypes at a given target, two haplotypes are chosen at random and the sample is treated as a heterozygote. It happens on average in less than 1% of the amplified targets that there are more than two different sequences, so we expect that this inexact way of dealing with those cases does not have a major impact on the results. Because we model the *8*-mer counts as the observations of a Poisson distribution, the counts have to be integers, hence we cannot average over all observed alleles as in the nearest neighbour method. If a given target did not amplify in the test sample, the associated *8*-mer count table will just contain zeroes, equivalent to simply ignoring missing data. To obtain the VAE input, we sum the *8*-mer tables over all 62 targets. This results in a single table per sample, with $65,536(=4^8)$ integer entries, roughly summing to twice the number of basepairs covered by the amplified targets, so a little under 20,000 for a sample in which all targets amplified.

## Variational autoencoder

The within-complex assignment is based around a variational autoencoder. We considered two other non-linear dimension reduction methods to perform the projection step: UMAP (*McInnes et al., 2018*) and t-SNE (*van der Maaten and Hinton, 2008*). t-SNE does not have the ability to embed new samples onto an already existing projection. This means that the distribution of the training set

samples depends on the test set we are projecting, which is not desirable for a classification problem. UMAP can distinguish between most species in the *An. gambiae* complex. The projections have a distinctly different quality from the VAE: the clusters are much tighter and well separated, there are no 'fuzzy' boundaries as in the VAE. However, these tight and well separated clusters come at the cost of containing some hard misclassifications; that is samples of one species, which lie well inside the cluster of a different species, which makes it hard to attach any measure of uncertainty to the assignments. For a full description of the UMAP projections and their comparisons with the VAE projection, see Appendix 2. We also ran ADMIXTURE on our data, but it did not separate the species in GCref v1 by assigning them to a unique ancestral population, see Appendix 3.

The VAE consists of an encoder, a latent space projection and a decoder. The specific design we used was inspired by popVAE (*Battey et al., 2021*). The encoder is a fully connected neural network that takes high-dimensional data as input and encodes that as a point in a latent space of much lower dimension. The decoder is also a fully connected neural network, and it takes as input a point in the latent space and outputs 'simulated' data of the same dimensions as the input data. The VAE learns a 'good' encoding by adjusting the weights in the encoder and decoder to obtain an output similar to the original input. To prevent overfitting, the input of the decoder is not the exact output of the encoder, but a nearby point in latent space. Furthermore, the loss function used to update the encoder and decoder weights contains a regularisation term on the latent space, in addition to the term measuring the similarity of the decoder output and the original input. Because of the introduced sampling noise and the regularisation constraint, the most efficient way to encode the data is to represent samples that are similar in the high-dimensional data by nearby points in the low-dimensional latent space. In summary, we expect that species identity shapes the structure in the *8*-mer count tables and that the VAE projects this structure to the low-dimensional latent space, resulting in clustering by species in the latent space.

In our case, the encoder input is the *8*-mer count table of the training set, so a table of dimension $n$ x 65536 with non-negative integer entries, where $n$ is the number of samples in the training set. The output of the encoder is a set of 2$d$ parameters for each sample, where $d$ is the dimension of the latent space. For each dimension, one parameter corresponds to the mean position in latent space and one parameter corresponds to the variance of the position in latent space. In our case, we use a three-dimensional latent space ($d$=3). The input of the decoder is a position in latent space for each sample, sampled from the distribution determined by the encoder output. The decoder output is an $n$ x 65536 dimensional table of strictly positive entries, however, unlike the input table, the entries are not necessarily integers.

The loss function is the sum of two terms: one measuring the difference between the input and output data and one acting as a regulariser on the latent space. The relative weight of these terms can be adjusted. If we model the count tables as observations of independent Poisson variables, the difference between the input and output can be measured as the Kullback-Leibler divergence (KL divergence) of the Poisson distribution with the means given by the output from the Poisson distribution with the means given by the input. The KL divergence is the same up to a constant as the negative of the Poisson loglikelihood with the input as the observed counts and the output as the means. So minimising the KL divergence is equivalent to maximising the loglikelihood with respect to the output. The difference term of the loss function is obtained by summing over all unique *8*-mers. The theoretical minimum of the difference term of the loss function is zero, and this is attained if the output is exactly the same as the input. However, this theoretical minimum cannot be attained in practice, because the sparsity of the input implies that there will be entries equalling zero and the activation function of the decoder generating the output results in strictly positive entries.

The regularisation term of the loss function is based on the KL divergence of the normal distribution parameterised by the encoder output from a standard normal distribution, that is $N(0,1)$. The regulariser is computed for each latent space dimension separately. Again, the theoretical minimum equals zero and is attained when the mean outputted by the encoder is zero and the variance outputted by the encoder is one. The regularisation term of the loss function is defined as the KL divergence summed over the latent space dimensions. This effectively enforces the distribution specified by the encoder output to look like a multi-dimensional Gaussian distribution with mean zero, variance one and covariance zero. In layman's terms, it pulls the projected positions of the samples in latent space towards the origin and establishes a natural scale for them. The regulariser prevents overfitting by

making it expensive for the encoder to place samples far away from the origin. The loss function used to train the VAE is the weighted sum of the similarity term and the regularisation term described above, with a parameter *w* that controls the relative strength of the two terms.

We set most of the parameters involved in training the VAE by comparing the latent-space projections for different parameter values, using a subset of GCref v1 containing only samples from *An. gambiae, An. coluzzii,* and *An. arabiensis.* The criteria we used to pick parameter values were species classification accuracy of the reference set and the validation set, using assignments based on convex hulls (described below), visible within-species structure, and, as a secondary criterion, useful visualisation. Further detail on the choice of parameter values is provided in Appendix 4. The latent space projection is also affected by the training dataset. We observed that the presence or absence of most countries does not affect the classification, except for the Gambia and Guinea-Bissau. When these countries are removed, the accuracy to distinguish between *An. gambiae* and *An. coluzzii* considerably reduces. This is not surprising, since these samples lie on the boundary of the *An. gambiae* and *An. coluzzii* clusters and as such are crucial for assigning those species. The complete results can be found in Appendix 5.

## Within-complex species classification

We use the trained and tuned VAE to assign species as follows. We input the summed *8*-mer table of the test samples into the encoder of the VAE. The encoder outputs a position in latent space for each sample. Importantly, the VAE is agnostic to species labels; the species assignment happens based on the position in latent space of the test samples in relation to the latent space positions of the species-labelled reference dataset GCref v1.

The top two panels of *Figure 4* show the latent space projection of GCref v1. While most species form nicely isolated clusters, *An. gambiae* and *An. coluzzii* border each other closely. Interestingly, the boundary is formed by mosquitoes from The Gambia and Guinea-Bissau. These mosquitoes are labelled as *An. gambiae* by conventional molecular barcoding, but they cannot be confidently assigned to either *An. gambiae* or *An. coluzzii* using over 500 ancestry informative markers (AIMs) or whole genome PCA (*Anopheles gambiae 1000 Genomes Consortium, 2017*; *Clarkson et al., 2020*). The clusters containing *An. bwambae* and the putative new species *An. fontenillei* are placed very close to each other, and can also not be reliably distinguished. These species are closely related, but up until now they have only been discovered in Uganda and Gabon, respectively, and so, since they do not seem to have overlapping geographic species ranges (*Barrón et al., 2019*), the species identity of samples falling into either of these two clusters can be resolved by their geographic origin.

We perform species classification using the convex hulls of species clusters. A convex hull is the mathematical notion of the smallest convex set containing all points of interest. A nice metaphor is to imagine that you are wrapping all points corresponding to samples of a single species together in such a way that requires the minimal amount of wrapping paper. We constructed one convex hull for each species represented in the dataset, using latent space positions of the samples from GCref v1 as well as of 363 additional species-labelled samples that were not used in training the VAE, to account for possible effects caused by projecting the samples to the latent space (sample information to be found in *Supplementary file 7*). For our classification procedure, it is important that the convex hulls of different species do not overlap. When constructing convex hulls from the full sample set, only the convex hulls corresponding to *An. gambiae* and *An. coluzzii* overlap. We trimmed these hulls by iteratively removing samples from the set of points used to construct them until they did not overlap. In total, we removed 17 *An. gambiae* and 6 *An. coluzzii* samples. The samples from *An. bwambae* and *An. fontenillei* are combined in one convex hull because they are so close together in latent space.

The classification of new samples happens as follows. If the latent space position of the test sample falls inside a convex hull, the sample is classified as that species. If the latent space position of the test sample falls outside all convex hulls, there are two options. If the sample is much closer to one convex hull than to all others, it is classified as the species corresponding to the hull it is closest to. To be precise, this happens if the euclidean distance to the closest convex hull is at least 7 times smaller than the distance to all other convex hulls. This allows for 'fuzzy' boundaries of the convex hulls that are proportional to the separation between the different hulls. We fitted the parameter value 7 on the dataset from Gabon to reflect the assumption that *An. tengrela* is not believed to be found in Gabon. If the latent space position of the test sample falls outside all convex hulls and outside their fuzzy

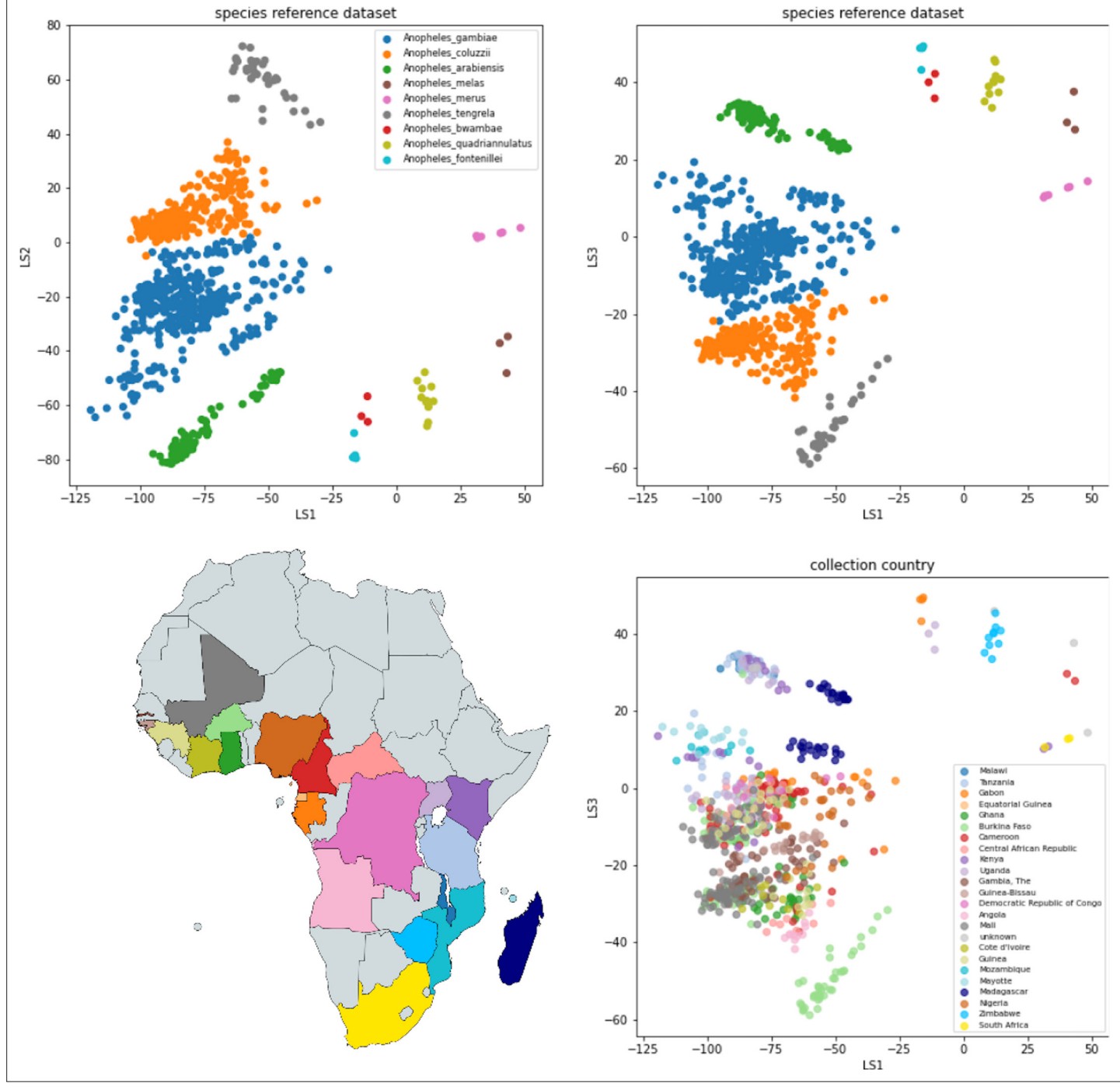

**Figure 4.** VAE projection of the *gambiae* complex reference dataset GCref v1. Top panels: the samples are represented by dots at the inferred mean position in three-dimensional latent space and coloured by their species label. The left panel shows latent dimension 1 versus latent dimension 2 and the right panel shows latent dimension 1 versus latent dimension 3. Bottom panel: the same projection as above, but here the samples are coloured by the country of collection revealing structure related to geography.

boundaries, the sample is assigned 'uncertain' followed by the labels of all the species whose convex hulls are within a radius of 7 times the distance to the closest convex hull, in order of proximity. This assignment reflects the uncertainty in our classification of samples that fall in an area in latent space where no species-labelled samples fall. At the same time, it gives information on our best guess or guesses for the species identity and leaves open the possibility to modify the assignments based on prior knowledge, for example species ranges or habitat restrictions.

There is one exception to the above assignment procedure: if the closest two convex hulls are from *An. gambiae* and *An. coluzzii* and the euclidean distance to both of these hulls is less than 14 then the test sample is classified as *uncertain_gambiae_coluzzii* or *uncertain_coluzzii_gambiae*, depending on which convex hull is closer, because we cannot reliably distinguish between these species in this part of the latent space.

In addition to structure driven by species, the latent space projection of GCref v1 also exhibits geographical structure. Within the *An. gambiae* species cluster, there is a distinct subcluster of samples from Madagascar. Similarly, there is a distinct subcluster of samples from Madagascar in the *An. arabiensis* species cluster. As mentioned before, the boundary between the *An. gambiae* and *An. coluzzii* species clusters is formed by samples from the far West of Africa. These samples also stand out as a separate group in a study on whole genome data (*Caputo et al., 2021*; *Clarkson et al., 2020*). There also appears to be a cluster of East African *An. gambiae* samples that are distinct from the main cluster of *An. gambiae* as well as the Madagascar samples. It is promising to see signatures of geographic structure within species from the amplicon panel data, because this suggests that the panel can also be useful to explore population structure within species.

## VAE classification accuracy

We applied the species assignment procedure to GCval v1 (*Figure 5*). 134 out of 142 samples (94.4%) samples are assigned to a single species and 132 of those (98.5%) are assigned to the species concordant with their species label. One sample labelled as *An. coluzzii* is classified as *An. gambiae* and one sample labelled as *An. gambiae* is classified as *An. coluzzii*. For all eight samples classified as uncertain, the reference species label is among the set of assigned labels. Seven of the samples classified as uncertain had fewer than 45 targets and we know that the proportion of missing targets affects the position in latent space of the projected samples. The other sample classified as uncertain falls in the space between *An. gambiae* and *An. coluzzii*. Further information can be found in *Supplementary file 6*.

## Case studies
### Ag1000G whole genome sequenced samples that are too diverged from the reference genome

The Ag1000G project removes samples from its analysis that appear not to be *An. gambiae*, *An. coluzzii,* or *An. arabiensis* based on their divergence from the PEST (*An. gambiae*) reference genome. We ran NNoVAE on all samples that fail the divergence filter from data releases v3 and v3.1 through v3.5. In these datasets, 212 of nearly 10,000 samples were filtered due to high divergence from the PEST reference genome; we assign 166 of those to *An. funestus* at the fine level and 17 to the *An. gambiae* complex at the intermediate level. Furthermore, we assign one sample to *An. nili gp sp3* and one to *An. jebudensis* at the fine level, and one to *An. marshallii* group at the intermediate level. There are 20 samples that get assigned only at the coarse level to the *Myzomyia Neocellia* series and six samples do not get assigned at any level. See *Supplementary file 8* for all sample and assignment information.

All 17 samples that are assigned to the *An. gambiae* complex had at least 50 targets. We assigned 5 samples to *An. merus*, 11 to *An. melas* and 1 to *Uncertain_melas_quadriannulatus* (*Figure 5*). The geographic origin of the samples assigned to *An. merus* and *An. melas* is compatible with the known ranges of these species (*Wiebe et al., 2017*).

Two of the unassigned samples stand out by their assignments: one sample appears to be from the *An. gambiae* complex but contaminated by an *An. funestus* sample; the second appears to be a member of the *Culex* genus rather than the *Anopheles* genus, based on mitochondrial analysis (data not provided). For this sample, we extracted only 13 targets. The remaining unassigned samples and those only assigned at the coarse level have at least 42 targets. They can be split into four groups of samples that are similar to each other in assignment proportions and mitochondrially. Because of the high amplicon recovery rate and the similarity of assignment proportions, often found in different countries, we believe that these samples represent species that are not present in NNv1. We aim to confirm the species identity for these groups by morphology and genomic comparison to publicly available data and, assuming this is successful, to include them in the next update of the reference database.

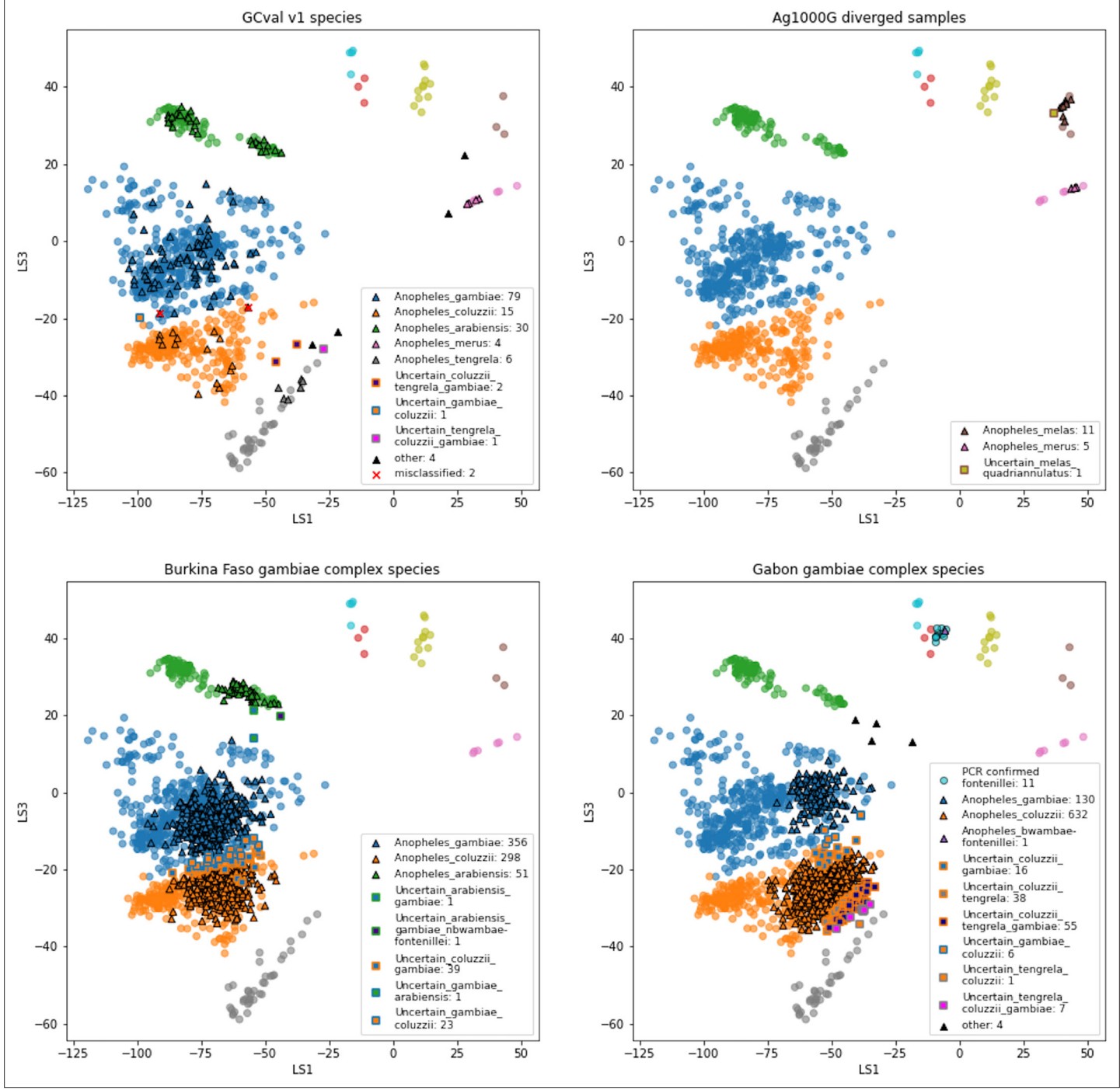

**Figure 5.** VAE projections of (**A**) validation set GCval v1 (**B**) diverged samples from Ag1000G (**C**) sample set from Burkina Faso (**D**) sample set from Gabon. The samples from the reference set GCref v1 are displayed as half transparent circles, coloured by species as in *Figure 4*. The samples from each of the projected sample sets are coloured by their assigned labels. The numbers behind each label corresponds to the number of samples in that category. Samples with more than 3 species-labels are listed as 'other'. The two samples from GCval v1 for which the species label does not match the assigned species are marked with a red cross.

## Burkina Faso

We collected 950 mosquitoes from three different locations in Burkina Faso. All individuals were morphologically assigned to the *An. gambiae* complex. For 905 individuals (95.3%) we obtained at least 10 amplified targets, the minimum number required for NN assignment. A total of 901 samples are indeed assigned to

the *An.gambiae complex* at the intermediate level, 2 samples are assigned to *An. nili group sp3* at the fine level, 1 sample could only be assigned at the coarse level to the *Myzomyia Neocellia* series and 1 sample could not be assigned at any level. The latter two samples require more detailed morphological investigation, which is made possible by the non-destructive extraction approach we used on these mosquitoes.

For 770 of the samples assigned to the *An. gambiae complex* (85.5%) we obtain at least 50 amplified targets, the minimum number required for VAE assignment. We find three species in this dataset: *An. gambiae, An. coluzzii,* and *An. arabiensis* (**Figure 5**). Most samples (91.6%) could be assigned to a single species, but we also find some samples labelled as uncertain shared between *An. gambiae* and one other species. The two samples that fall in the space between the *An. gambiae* and *An. arabiensis* reference samples are remarkably far away from all other samples. We cannot exclude the possibility that contamination between samples plays a role here, especially because these samples were stored with 10 individuals in ethanol in a single 1.5 mL tube in the years before sequencing. All metadata and assignment information can be found in **Supplementary file 9**.

## Gabon

We collected 1,056 mosquitoes by human landing catch from different locations in Lopé village in Gabon. All individuals were morphologically identified as members of the *An. gambiae* complex. Eleven mosquitoes were identified as *An. fontenillei* by a species diagnostic PCR (**Fanello et al., 2002**) and those clustered with the *An. fontenillei* and *An. bwambae* samples in the VAE projection as expected. Those samples were used to construct the convex hulls. For 1002 (95.9%) of the unidentified samples we obtain at least 10 targets and these are classified by the NN step of NNoVAE. The vast majority (993 samples; 99.1%) is assigned to the *An. gambiae complex*. The other species we find in this dataset are *An. funestus* (two individuals) and *An. coustani complex* (two individuals), plus five individuals that could not be assigned at any level. The five unassigned individuals had a high proportion of multi-allelic targets, suggesting contamination may have occurred at some point in the process.

Of the 993 mosquitoes assigned to the *An. gambiae* complex, 890 (89.6%) have sufficiently many targets to be run through the VAE (**Figure 5**). Most of these are assigned to a single species: 632 to *An. coluzzii,* 130 *An. gambiae,* and 1 *An. fontenillei* (the latter is actually assigned to the combined convex hull of *An. fontenillei* and *An. bwambae*, but are assumed to be *An. fontenillei* because of the non-overlapping species ranges). *An. fontenillei* was first discovered in the forest of La Lopé National Park, 10–15 km south of the sampling locations presented in this study (**Barrón et al., 2019**). Although the species strongly prefers forested habitats, it is not surprising to find one of them in the village given the small distance. It is noticeable that a large number of samples fall in between the *An. coluzzii* and *An. tengrela* clusters. Judging from the projection, we suspect that the whole cluster is *An. coluzzii,* but it may be interesting to explore what drives the variation within this large cluster and why so many samples are projected to an area where none of our reference samples are. Four other samples are projected close to the *An. arabiensis* cluster, although they are assigned to a large set of species labels. Independent PCR species diagnostics confirmed these four samples as *An. arabiensis* (**Fanello et al., 2002**). This was surprising because *An. arabiensis* has not been observed in Gabon before, despite extensive sampling efforts. The fact that we see only four such specimens in this set of more than 1,000 with sufficient data, demonstrates that whatever species these mosquitoes are, they are probably quite rare or less likely to be caught by human landing catches. We plan to generate whole genome sequencing data for these individuals to investigate their species and their relationship to *An. arabiensis* from other geographic locations and to other sympatric species. All sample information and assignment results are listed in **Supplementary file 10**.

## Discussion

In this paper we presented NNoVAE, a method for robust species identification for the entire *Anopheles* genus from multilocus targeted amplicon sequencing data. This integrated approach removes the need for sorting the specimens into species groups or complexes based on morphology, which is labour intensive and error prone, particularly in the case of damaged mosquitoes, for instance when collected with CDC light traps. The NN step can distinguish between most species in our reference database. But equally importantly, it gives an indication of the uncertainty of the assignment, using the same thresholds for the

entire genus, enabling us to quantify confidence in assignment in a meaningful way allowing for comparison between all species groups. For individuals from species not yet represented in the reference dataset, we can often assign to a species-group at the intermediate level (corresponding to taxonomic species groups or complexes) or at the coarse level (corresponding to taxonomic series or subgenera). A few samples did not meet the threshold to be assigned at the coarse level, but the assignment results do give an indication of the position of these samples in the phylogeny. Initial explorations of the mitochondrial sequences for those samples do not indicate a close match to publicly available mitochondrial data of any *Anopheles* species. We hope to resolve the species identity of these samples by extending our reference database and collaborating with morphological experts, but until then we retain the groups of unresolved species to compare future samples against them.

The NN step alone struggles to differentiate closely related species within species complexes. For the *An. gambiae* complex we developed a high resolution species identification method based on a variational autoencoder. This VAE step should be easily extendable to other species complexes for which a sufficient amount of species-labelled data is available, which we expect to be the case soon for *An. funestus* and *An. coustani*. The VAE can accurately distinguish between eight species in the *An. gambiae* complex; only *An. bwambae* and *An. fontenillei* are too close together in the projection to reliably separate them, but geographic origin helps with this. The species assignments for the VAE are currently quite conservative; only if the VAE projection of the test sample falls within the cloud of training samples from a single species, or much more close to it than to any other cluster, is it assigned to that species. Otherwise it gets assigned all the species labels of nearby clusters. This way, we flag potential outliers or unexpected species, as for the *An. arabiensis* in Gabon, but entomologists can still decide to exclude certain species labels if they are sure that they are not appropriate for their collection location and time, for instance the *An. tengrela* label in Gabon.

The VAE projection of the *An. gambiae* complex also shows some population structure within species clusters. Some of the structure reflects the geography of the collection locations, for example Madagascar stands out as a separate subcluster both for *An. gambiae* and *An. arabiensis*. However, the sample sets from Burkina Faso and Gabon show that samples from the same location can be projected to different positions in latent space and it would be interesting to investigate what is driving this observed diversity.

NNoVAE relies on a reference database and therefore the accuracy of the species assignments also depends on the quality and completeness of the reference database. We expect that version 2 of the reference database will contain approximately fifty additional species, as well as more individuals of species that are underrepresented in NNv1 and we are seeking further well-characterised samples.

Our goal over the next 2 years is to adapt the ANOSPP protocol, which currently uses high-throughput robotics equipment at Sanger, to run it in any basic molecular laboratory. We plan to build an accompanying website where sequence data will be automatically accessible and data analyses can be run interactively and openly (if desired). We hope that the combination of the complete end-to-end protocol from sample to sequence interpretation together with a one stop shop for data interpretation will help connect vector control initiatives across the globe.

We believe that this genomic species identification tool for the entire genus is extremely valuable to monitor *Anopheles* populations at large scale. We think that similar tools could benefit researchers studying other genera with morphologically similar species, in particular when knowing the species identity is of medical importance. We hope that the methods outlined in *Makunin et al., 2022* and in this manuscript can be of help in developing a targeted amplicon panel for other important vector genera such as *Aedes* and *Culex* and that by extending NNoVAE as described here it will be straightforward to analyse data from such future panels.

NNoVAE can characterise vector populations in a uniform way across the *Anopheles* genus and as such contributes to our understanding of *Anopheles* species composition, population structure, species ranges, and transmission potential. We believe that large scale monitoring of *Anopheles* populations of all species is of pivotal importance for successful implementation of malaria control strategies. ANOSPP and NNoVAE enable us to catalogue species ranges and distributions and identify and record which species are found to carry *Plasmodium*. It has been shown that by successfully targeting a single vector species, a different species can become the main transmitter of malaria (*Okumu and Finda, 2021*). *An. stephensi*, an efficient vector species originally found in South East Asia and the Arabian peninsula, has been reported to spread in the Horn of Africa and was implicated in malaria outbreaks in Djibouti (*Faulde et al., 2014*; *Seyfarth et al., 2019*; *Ahmed*

*et al., 2021*). These examples underscore the importance of monitoring species and their transmission potential agnostically. ANOSPP and NNoVAE also provide an opportunity to study changes in species composition, for instance by comparing catches before and after implementing a malaria control intervention, or by comparing catches collected in different seasons or in areas undergoing land use changes. Lastly, ANOSPP and NNoVAE provide a quick and cheap way to select interesting samples (e.g. unexpected species for the collection area or season, or simply balanced numbers of different species) for whole genome sequencing to study important genomic features, such as evidence of population bottlenecks or spread of strongly selected putative insecticide resistance alleles. In conclusion, the combination of ANOSPP and NNoVAE offers a cheaper, more robust, more informative, and more reliable way to carry out malaria vector surveillance that we hope will be embraced over the coming years by the medical entomology community and National Malaria Control Programs.

# Methods
## Data processing
### Panel sequences
The amplicon sequencing data are processed as described previously (*Makunin et al., 2022*). In brief, the fastq files containing the reads are split into one file per target by cutadapt v2.5 (*Martin, 2011*), which uses the primer sequences to do so. Cutadapt also filters for read pairs where both the forward and reverse read match the appropriate primer and trims the primers. Next, DADA2 v1.10 (*Callahan et al., 2016*) is used to reconstruct sample haplotypes from these read pairs and they are filtered using a custom script to include only haplotypes supported by at least 10 read pairs and with at least 0.1 haplotype frequency per sample-amplicon pair.

### Reference genomes
In silico amplicon extraction from reference genomes is done as described previously (*Makunin et al., 2022*). Targets are extracted by matching the primer sequences in the reference genome using Seek-Deep v2.6 command 'genTargetInfoFromGenomes' (*Hathaway et al., 2018*).

### Publicly available data
Where short read whole genome sequence data are publicly available for *Anopheles* mosquitoes, we use these data to pull out target haplotypes to add to our reference index. If the genomic coordinates of the primers are known (e.g. the reads are aligned to a publicly available reference genome), we extract the reads overlapping the primer or target sites for each amplicon separately from BAM files and convert these to fastq files, using samtools v1.9 (*Li et al., 2009*). These are used as input for fermi-lite (*Li, 2015*), which creates an assembly graph. The unitigs from the assembly graph are cleaned up by cutadapt v3.1 (*Martin, 2011*): the sequences outside the primer sites are trimmed, while the sequences matching the primers are retained and the unitigs are oriented according to the primers. Next, the unitigs are merged using the information from the assembly graph with a custom python script, which relies on MAFFT v7.475 (*Katoh and Standley, 2013*) for sequence alignment. In the final step, the primers are trimmed from the resulting haplotypes by cutadapt and any haplotypes that do not have primer sequences on both ends are removed, which helps to get rid of contamination. If the genomic coordinates of the primers are not known, we align the samples to the most appropriate reference genome, identify the genomic coordinates of the primer sites if not yet known and follow the steps above.This pipeline is implemented in Snakemake 5.30.2 (*Mölder et al., 2021*). Snakefile and scripts are available on GitHub.

## Data structure
The resulting haplotypes are stored in a table with columns recording the sample name, target, haplotype sequence, read count of supporting reads (for amplicon data only), fraction of supporting reads (for amplicon data only). So each row corresponds to a unique haplotype for a sample target combination, hence samples that are heterozygous at a certain target will have two rows for the same target.

## Implementation

Species labels are assessed and species-groups constructed using custom Python scripts implemented in python 3.8 (**Van Rossum and Drake, 2009**). The VAE is implemented in keras 2.3 (**Chollet, 2015**) using a custom python script. Convex hull construction and distance computations rely on scipy 1.6 (**Virtanen et al., 2020**) and pygel3d 0.2 (**Baerentzen, 2018**). Plots are created with matplotlib 3.3 (**Hunter, 2007**) and seaborn 0.11 (**Waskom, 2021**). All scripts and environments are available on GitHub.

## *K*-mers

Alignments of amplicon target sequences from highly diverged species are often poor and it is difficult to define a 'fair' distance metric based on these alignments. Moreover, there is not a straightforward way to account for small indels and structural variants with alignment based distances. *K*-mer based distances naturally incorporate indels and structural variation and account for highly diverged sequences in an objective way and provide a solution to the problems arising from relying on alignments. Therefore, our species assignment method uses *k*-mers to support better comparisons between the sequences in the database and the sequences of the test sample.

There is a trade-off in the choice of *k*. For large *k* there is little tolerance for errors, while for small *k* there is a high chance that the same *k*-mer is found in multiple locations in the sequence. For example, in a 149 bp sequence, 5 evenly spread SNPs result in no 25-mers matching the reference. On the other hand, the chance that all 4-mers are unique in a sequence of the same length is incredibly small ($<10^{-22}$). Based on these trade-offs, we selected 8-mers as a reasonable length. The total sequence length of the amplicon panel targets for the current *An. gambiae* PEST reference genome sequence AgamP4 is 9928 bp, with a mean target length of 160 bp. There are 65536 unique 8-mers, so the chance that all 8-mers within a target are unique is 84% on average. Across the nearly 10 kb of amplified sequence, the chance that all 8-mers are unique is vanishingly small, but the expected number of unique 8-mers is approximately 8533 (sd 46) and the expected number of non-unique 8-mers is approximately 680 (sd 22).

The methods we present here work with *k*-mer tables created from each haplotype from each target. A *k*-mer table consists of $4^k$ columns, each corresponding to a unique *k*-mer. To translate a sequence to a *k*-mer table, we record in each column how often the corresponding *k*-mer occurs in the sequence. This results in a sparse table (the sparsity of course depends on the choice of *k*) with non-negative integer entries. As an example, the *2*-mer table for the sequences AACTACTCT (first row) and AGCTACTT (second row) is shown below. Note that *all* possible *k*-mers are represented in the table, even when they do not appear in any sequence.

| AA | AC | AG | AT | CA | CC | CG | CT | GA | GC | GG | GT | TA | TC | TG | TT |
|----|----|----|----|----|----|----|----|----|----|----|----|----|----|----|----|
| 1  | 2  | 0  | 0  | 0  | 0  | 0  | 3  | 0  | 0  | 0  | 0  | 1  | 1  | 0  | 0  |
| 0  | 1  | 1  | 0  | 0  | 0  | 0  | 2  | 0  | 1  | 0  | 0  | 1  | 0  | 0  | 1  |

The *k*-mer distance between two sequences is defined as follows. Translate both sequences to *k*-mer tables as described above and call these $q_1$ and $q_2$. Then the *k*-mer distance between them is given by $d_k(q_1, q_2) = \frac{\Sigma |q_1 - q_2|}{\Sigma |q_1 + q_2|}$, that is the number of non-matching *k*-mers divided by the total number of *k*-mers in both sequences. The normalisation is required to correct for a bias attributing smaller distances to shorter sequences. In the nearest neighbour method we use this *k*-mer distance to compare one haplotype sequence of a test individual against one haplotype sequence from the reference database. Note that this definition relies on the *k*-mer counts, not simply on the presence or absence of each *k*-mer. Using this definition, the *2*-mer distance between AACTACTCT and AGCTACTT is 7/15.

## Data collection and availability

All mosquitoes collected as part of this work were done so under the local guidelines and with permission, which vary depending on the country of origin. ABS (Nagoya) compliance for receiving mosquitoes or their DNA in the UK was completed for all mosquitoes used in this research. Raw sequencing data will be made available on ENA (accession to be confirmed). Pipelines and analysis code, together with processed target haplotypes are available on GitHub: https://github.com/marilouobodde/NNoVAE.

## Acknowledgements

We thank Vickie Brooks for general logistic support for ANOSPP. We thank Petra Korlević for support in wet lab work, valuable discussions and conducting the mitochondrial analysis. For field sampling in Burkina Faso, we would like to thank Franck Adama Yao, Patric Stéphane Epopa Ngomé, Nouhoun Traoré, and the Target Malaria Team. We thank Samuel Anifowose, Akinkunle Adeniyi, Joshua Oduwu, Idris Otun, Juwon Adeniji, Wale Enisemo, Cynthia Umunnakwe and Adedotun Bayegun for sample collection and processing in Nigeria. In Gabon, we thank the ESV team at the CIRMF. We also thank the Institut Pasteur de Madagascar for sample collection in this country. We thank Sanger's Scientific Operation Teams for carrying out all PCRs, library generation, and sequencing on data presented here and Catherine McCarthy for her support in ensuring all samples are compliant with Access and Benefit Sharing of sequence data as laid out by the Nagoya Protocol. The Wellcome Sanger Institute is funded by the Wellcome Trust (206194/Z/17/Z), which supports MKNL and the work contained here. MB was supported by Wellcome 4 year PhD studentship (RG92770), and RD by Wellcome award (WT207492). Gabon collections were supported by an ANR grant to DA (ANR-18-CE35-0002-01 – WILDING). LB was supported by an IRD student fellowship (Bourse ARTS/IRD).

## Additional information

### Funding

| Funder | Grant reference number | Author |
| --- | --- | --- |
| Wellcome Trust | 206194/Z/17/Z | Mara KN Lawniczak |
| Wellcome Trust | RG92770 | Marilou Boddé |
| Wellcome Trust | WT207492 | Richard Durbin |
| Agence Nationale de la Recherche | ANR-18-CE35-0002-01 - WILDING). | Diego Ayala |
| Institut de Recherche pour le Développement | Bourse ARTS/IRD | Lemonde Bouafou |

The funders had no role in study design, data collection and interpretation, or the decision to submit the work for publication. For the purpose of Open Access, the authors have applied a CC BY public copyright license to any Author Accepted Manuscript version arising from this submission.

### Author contributions

Marilou Boddé, Resources, Writing – review and editing, Conceptualization, Data curation, Formal analysis, Investigation, Visualization; Alex Makunin, Writing – review and editing, Methodology, Writing - original draft; Diego Ayala, Frederic Tripet, Supervision, Funding acquisition, Writing - original draft; Lemonde Bouafou, Mahamadi Kientega, Boris K Makanga, Supervision, Writing - original draft; Abdoulaye Diabaté, Uwem Friday Ekpo, Supervision, Funding acquisition; Gilbert Le Goff, Marc F Ngangue, Olaitan Olamide Omitola, Nil Rahola, Supervision; Richard Durbin, Mara KN Lawniczak, Resources, Methodology, Funding acquisition, Writing - original draft

### Author ORCIDs

Marilou Boddé http://orcid.org/0000-0003-2107-5100
Diego Ayala http://orcid.org/0000-0003-4726-580X
Uwem Friday Ekpo http://orcid.org/0000-0002-0543-5463
Olaitan Olamide Omitola http://orcid.org/0000-0003-3827-6320
Nil Rahola http://orcid.org/0000-0003-4067-6438
Mara KN Lawniczak http://orcid.org/0000-0002-3006-2080

### Decision letter and Author response

Decision letter https://doi.org/10.7554/eLife.78775.sa1
Author response https://doi.org/10.7554/eLife.78775.sa2

## Additional files

### Supplementary files

- Supplementary file 1. NNv1 metadata and species labelling information.
- Supplementary file 2. NNv1 coarse level assignment proportions.
- Supplementary file 3. NNv1 intermediate level assignment proportions.
- Supplementary file 4. NNv1 fine level assignment proportions.
- Supplementary file 5. GCref v1 metadata and VAE projection coordinates.
- Supplementary file 6. GCval v1 metadata and assignment results.
- Supplementary file 7. Additional samples for convex hull definitions, metadata and VAE projection coordinates.
- Supplementary file 8. Ag1000G diverged samples metadata and assignment results.
- Supplementary file 9. Burkina Faso metadata and assignment results.
- Supplementary file 10. Gabon metadata and assignment results.
- MDAR checklist

### Data availability

Pipelines and analysis code, together with processed target haplotypes are available on GitHub: https://github.com/mariloubodde/NNoVAE, (copy archived at swh:1:rev:560351c897476e4e869f-eafaa39a077fa5a7aa86). Raw sequencing data has been made available on ENA; accesions can be found on GitHub.

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

## Appendix 1

## Species labels

The wild-caught mosquitoes in the reference database were labelled either morphologically or confirmed with species diagnostic PCR. Species identities were further determined by Sanger sequencing the molecular barcodes Cytochrome *c* oxidase I (COI) and Internal transcribed spacer 2 (ITS2). Both markers were compared to the NCBI database to assess species identity, and COI was also compared to the BOLD database. (See *Makunin et al., 2022* for more information).

Some partner labels were revised when distances between the samples in the reference dataset suggested mislabelling. Preferably the relabelling was further supported by molecular barcodes. Relabelled samples are:

- Amar-5: *An. marshallii* → *An. gambiae*. The average distance to samples in the *An. marshallii* complex is 0.54 (0.52–0.58), while the average distance to samples in *An. gambiae/coluzzii* is 0.09 (0.07–0.13). Additionally, both COI and ITS2 have *An. gambiae* as best hit.

- Amar-42: *An. marshallii* → *An. jebudensis*. The average distance to samples in the *An. marshallii* complex is 0.38 (0.35–0.42), while the distance to *An. jebudensis* is 0.05. Unfortunately, neither *An. marshallii* nor *An. jebudensis* is present in the barcode databases.

- Amar-3–1: *An. marshallii* → *An_marshallii_cp_sp1*. It has the lowest distances to samples of *An. hancocki, An. brohieri, An. demeilloni, An. theileri*, on average 0.26 (0.22–0.29), while the distances of *An. hancocki, An. brohieri* and *An. demeilloni* samples to each other is on average 0.03 (0.02–0.04) and the distances of the *An. theileri* samples is 0.07 (0.05–0.08). The average distance between the group containing *An. hancocki, An. brohieri* and *An. demeilloni* and the *An. theileri* is 0.29 (0.25–0.33). So it seems like *Amar-3–1* does not belong to either of these two groups, but it is as related to these groups as they are to each other. Unfortunately it does not have any informative matches on the molecular barcodes. Therefore this sample is labelled as an unnamed species in the *An. marshallii* complex.

- Adem-15: *An. demeilloni* → *Myzomyia_sp1*. Its lowest distance to another sample in the database is 0.39, to an *An. funestus*. This is a much higher distance than a sample usually has to other samples of the same or of a closely related species. In particular, the distance of *An. hancocki, An. brohieri* and *An. demeilloni* samples to each other is 0.03 (0.02–0.04), while the average distance of Adem-15 to these samples is 0.41 (0.39–0.44). As it does not show similarity to any of the samples in the database, but it is closer to almost all samples in the *Myzomyia* series than to almost all samples outside the *Myzomyia* series, it is labelled as an unnamed species in the *Myzomyia* series.

- Acol-645: *An. coluzzii* → *An_gambiae_cp_sp1*. This sample is different from other *An. coluzzii* and *An. gambiae*, this was found both in the sample-pair differences and in the subsequent VAE method which is tailored at the *An. gambiae* complex. The molecular barcodes also suggest that this sample is not *An. coluzzii* or *An. gambiae*, but that it is a member of the complex. Therefore, it is labelled as an unnamed species in the *An. gambiae* complex.

- Apal-257: *An. paludis* → *An_coustani_cp_cl3*. Its smallest distances are to samples of *An. tenebrosus, An. ziemanni, An. coustani* and *An. paludis*, on average 0.29 (0.28–0.32). We found that the samples of these species split into two clades and the distance of Apal-257 to either of these clades was much higher than the distances between samples in the same clade. Additionally, the average distances between the two clades is 0.18 (0.14–0.21), so Apal-257 is also further away from the two clades than they are from each other. Yet, it is closer to these two clades than it is to *An. sinensis* and *An. hyrcanus*, which are the next closest species. Regarding the molecular barcodes, Apal-257 matches to *An. coustani* on COI, like most samples in the *An. coustani* complex. On ITS2 it matches to *An. junlianensis* and *An. yatsushiroensis*, species in the hyrcanus group. We therefore decided to relabel it to a third clade in the *An. coustani* group.

- *An. nili* samples. Anils-7: *An. nili s.s.* → *An_nili_gp_sp1*. Anil-237 & Anil-239: *An. nili* → *An_nili_gp_sp2*. Anil-233, Anil-236 & Anil-238: *An. nili* → *An_nili_gp_sp3*. Both the distances and alignment of the molecular barcodes support this split. It is possible that some of these samples represent different member species of the *An. nili* group, which have been shown from cytogenetic analysis to differ substantially (*Sharakhova et al., 2013*), but molecular barcodes are not yet available in public databases for all member species.

| dist | An_nili_gp_sp2 | An_nili_gp_sp3 |
|------|----------------|----------------|
| An_nili_gp_sp1 | 0.19 (0.19–0.20) | 0.24 (0.23–0.26) |
| An_nili_gp_sp2 | 0.02 | 0.24 (0.24–0.26) |
| An_nili_gp_sp3 | | 0.04 (0.03–0.05) |

- *An. hyrcanus* samples. VBS00085 & VBS00086: *An. hyrcanus* → *An_hyrcanus_gp_sp1*. VBS00082 & VBS00083: *An. hyrcanus* → *An_hyrcanus_gp_sp2*. The distances within these pairs are 0.08 for both pairs. The distances between the pairs are 0.35 (0.34–0.36). The molecular barcode matches support the split. Even though hyrcanus is present in all databases, there are few matches to it. On COI in both BOLD and NCBI, the first pair matches to *nitidus* and the second pair to *crawfordi*; these species are in two distinct subgroups of the hyrcanus group. On ITS2 the first pair matches to *hyrcanus* and then *nitidus*, and the second pair matches to *sinensis* (*sinensis* is in the main *hyrcanus* group, not in either of the aforementioned subgroups). The alignments of COI and ITS2 sequences for these four samples also clearly support the split into pairs.

- Samples in the *An. coustani* complex. This complex contains the species *An. tenebrosus, An. ziemanni, An. coustani* and *An. paludis*. The between sample distances indicate a split into two clades (disregarding sample Apal-257, discussed above); however, the split is not correlated with the species labels. The proposed clades are *An_coustani_cp_cl1* containing Aten-191, Aten-185, Azie-334, Acou-956, Acou-959, Acou-962 and *An_coustani_cp_cl2* containing Aten-333, Aten-79, Aten-954, Azie-1032, Azie-1055, Azie-70, Azie-77, Acou-71, Acou-80, Apal-81. The average distances between members of the same clade is 0.07 (0.05–0.09) and 0.07 (0.05–0.10) respectively, while the distance between members of different clades is 0.18 (0.14–0.21). The barcode matches are to *An. coustani* for the vast majority of the samples, even though COI sequences for *An. ziemanni* and *An. tenebrosus* are present in both BOLD and NCBI database. Alignment of the COI sequences shows extremely little variation. Alignment of ITS2 does support a split into the two proposed clades.

All labels and groupings are displayed in ***Supplementary file 1***. In the majority of cases, the fine species-groups are supported by at least one molecular marker and not contradicted by different species labels. Some species are not present in one or both databases, so for these a match to a closely related species is allowed. The exceptions are:

- *An. jebudensis*: one sample was labelled as *An. jebudensis*, the other one originally as *An. marshallii*, but showed sufficient evidence for relabelling. Neither *An. jebudensis* nor *An. marshallii* is present in either database. On ITS2 they both match to *An. moucheti*, which is the closest species in the dataset, neither has a match in BOLD, and on COI in the NCBI database the best matches are to *An. lindesayi* and *An. sawyeri*, which are in the *Anopheles* and *Nyssorhynchus* subgenus respectively. However, it has to be noted that the NCBI database does not contain an *An. moucheti* COI sequence and the best hits are based on 81% and 91% identity respectively.

- *An. rhodesiensis*: All samples match to uninformative 'Anopheles_sp'; even though a COI sequence is available in BOLD and NCBI. The best hit to a named species in the dataset is to *An. funestus* for COI and *An. aconitus* for ITS2.

- *An. jamesii*: Is predicted for BOLD and ITS2, but not for COI in the NCBI database, even though an *An. jamesii* COI sequence is present there.

- *An. maculatus A*: has a few matches to *An. sawadwongporni*, which is also a member of the maculatus group, even though sequence of *An. maculatus A* is present in all databases.

- *An. rampae*: the four different samples match to *An. rampae(2 x), An. sawadwongporni(1 x)* and *An. maculatus(1 x)* in the BOLD database; all these are in the same species group. In the NCBI database, all COI hits are for *An. maculatus* (*An. rampae* COI not present in database), and all ITS2 hits are *An. rampae* and the uninformative *Anopheles_sp*.

- *An. balabacensis*: matches to *An. introlatus* on COI in both databases, even though *An. balabacensis* sequence is present. The two species are in the same complex.

- *An. carnevalei*: all samples match to *An. carnevalei* on BOLD and ITS2. *An. carnevalei* sequence is not present in NCBI database for COI; four out of five samples match to *An. nili*, one to *An. darlingi*, which is in a different subgenus.

- *An. dureni*: sequence not present in either database; on ITS2 one sample matches to *An. minimus* and one to *An. leesoni*, which are both in different groups; but placement of *An. dureni* is not very clear on the species tree.
- *An. vinckei*: sequence not present in either database; on COI there are distant matches to *An. gambiae* and *An. maculatus*, which are both in different groups; but placement of *An. vinckei* is not very clear on the species tree (seems to be close to *An. dureni*).
- *An. coustani* group: we have samples of four species, *An. coustani* (present in all databases), *An. paludis* (present in no databases), *An. ziemanni* (COI present in both databases), and *An. tenebrosus* (COI present in both databases). In BOLD, all samples match to *An. coustani*. In the NCBI database, on COI most matches are to *An. coustani*, there is one match to *An. ziemanni*, and a few matches to *Anopheles_cf*. On ITS2 there are many matches to *Anopheles_sp*. and *Anopheles_cf.*, and the remaining matches are to *An. coustani* and one sample to *An. junli-anensis* and *An. yatsushiroensis* in the hyrcanus group (this is the *An. paludis* sample, that is further removed from all other samples in this complex). These samples also show a 'checker-board' pattern, which splits them into two clades, which are not correlated with species labels.
- *An. barbirostris*: there are matches to *An. barbirostris* and to the closely related *An. dissidens*. However, the samples do not form two separate groups based on the distances.
- *An. oryzalimnetes*: no Sanger sequencing done.
- *An. cruzii:* no Sanger sequencing done.
- *An. bellator*: no Sanger sequencing done.

Lastly, there are samples which are further than 0.10 distance away from other samples in the reference database representing the same species. Above we have discussed some samples where there was good evidence to adjust the species label, but there are also some for which there is good reason to retain the partner label. Those are:

- VBS00001: *An. annularis*. It is a bit more diverged from the other three *An. annularis* samples, which causes it to fall just above the threshold. However, the distances are not very different from the distance between the other *An. annularis*, the next closest sample in the reference set are much further away and the molecular barcodes support the partner label.
- Agam-37: *An. gambiae*. It is a bit more diverged from the other *An. gambiae* that were sequenced by the panel, which causes it to fall just above the threshold. But is it closer to *An. gambiae* than to other samples in the reference dataset. The molecular barcodes also support the partner label.
- VBS00149 and VBS00150: *An. tessellatus*. They are 0.11 distance from each other, but for both the molecular barcodes match to *An. tessellatus* and the next closest sample in the database is at 0.44 distance.
- anopheles-sinensis-chinascaffoldsasinc2 & anopheles-sinensis-sinensisscaffoldsasinc2: *An. sinensis*. They are 0.11 distance from each other and that is still clearly closer than the next best match in their sister species *An. hyrcanus* (namely *An_hyrcanus_gp_sp2*).

## Appendix 2

### Comparison UMAP and VAE

We projected the *An. gambiae* complex training set (GCref v1) with UMAP, using the same 8-mer count tables as for the VAE as input. Using the default UMAP parameters does a very poor job at distinguishing species (*Appendix 2—figure 1*). By experimenting with different parameter values and different metrics, the three best-performing projections (judging by eye after colouring samples by their species label, see *Appendix 2—figure 1*) are:

- metric = hamming, n_neighbors = 3, min_dist = 0, n_components = 2
- metric = canberra, n_neighbors = 3, min_dist = 0.1, n_components = 3
- metric = braycurtis, n_neighbors = 3, min_dist = 0.99, n_components = 3

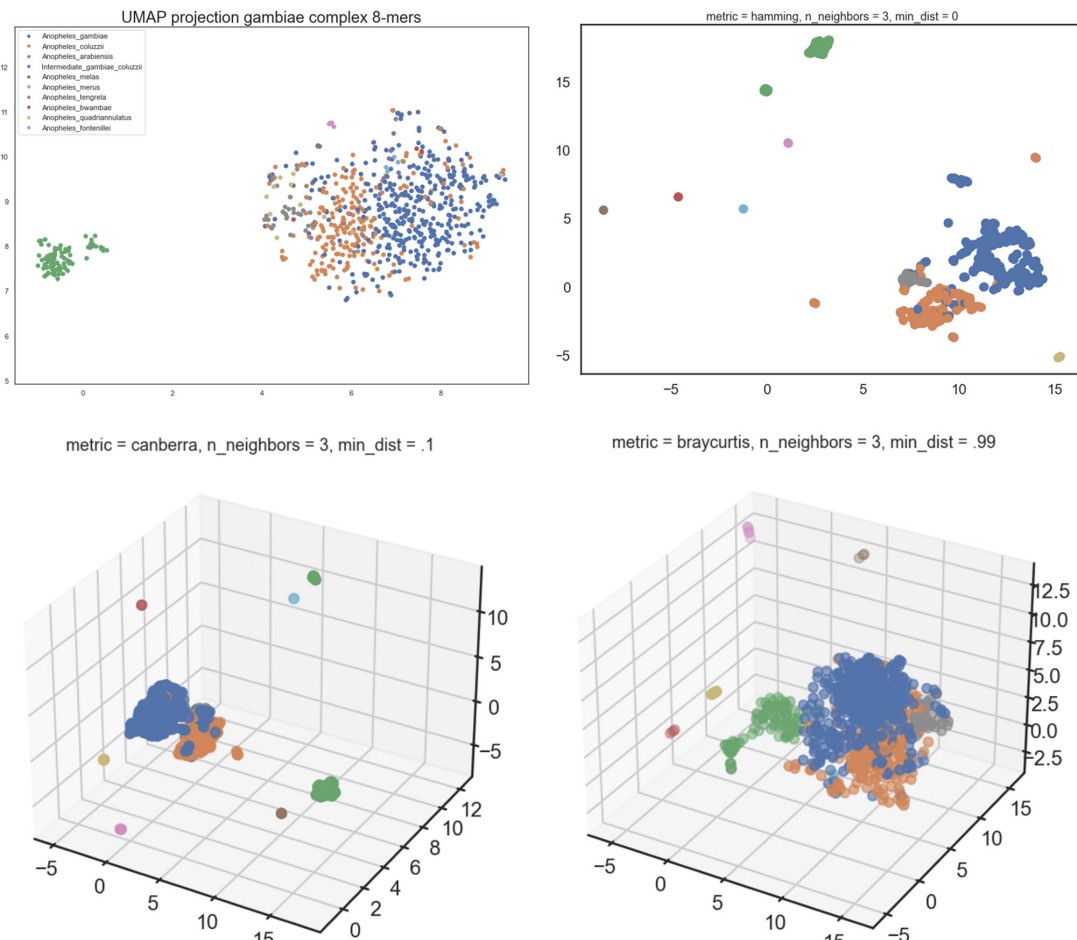

**Appendix 2—figure 1.** UMAP run on *8*-mer count table of *An. gambiae* complex dataset GCref v1. Left top is using the default UMAP settings, for the other three projections, the settings are specified in the title. UMAP was run unsupervised; the samples are coloured by their species labels after projecting them.

We have two species that are only represented by three samples and by setting the parameter n_neighbours higher than 3, these species will not split out as separate clusters. Although most of the species form distinct clusters in these three projections, there is still significant overlap between *An. gambiae*, *An. coluzzii* and *An. tengrela*.

It is noticeable that UMAP tends to generate tighter clusters than the VAE (see *Figure 4* in the main text); there are less fuzzy boundaries, but it comes at the cost of some samples being placed in the wrong cluster. The convex hull assignment we used with the VAE projection is specifically aimed to deal with the fuzzy boundaries and it doesn't cope well with outlier samples. Therefore, we use a different classification method, namely support vector machines, to perform species assignment

using the UMAP projections. We trained an SVM with the default parameters on the reference dataset GCref v1. The accuracy results and comparison are shown in the table below.

| | UMAP hamming | UMAP canberra | UMAP braycurtis | VAE |
|---|---|---|---|---|
| GCref v1 - correct | 93.3% | 94.6% | 96.3% | 98.5% |
| GCref v1 - uncertain | 0% | 0% | 0% | 2.4%* |
| GCref v1 - incorrect | 6.7% | 5.4% | 3.7% | 0.0% |
| GCval v1 - correct | 97.9% | 97.9% | 97.9% | 93.0% |
| GCval v1 - uncertain | 0% | 0% | 0% | 5.6% |
| GCval v1 - incorrect | 2.1% | 2.1% | 2.1% | 1.4% |

*this includes 0.9% of samples assigned to *Anopheles_bwambae_fontenillei*, which forms a combined assignment category in the VAE classification, and hence cannot be assigned to a single species.

We suspected we could improve the results by treating separate clusters of the same species as their own assignment category, for example relabel the samples in the separate *An. arabiensis* cluster as An_arabiensis-2. However, this did not improve the accuracy, neither for the reference set GCref v1 nor for the validation set GCval v1.

Overall the VAE has a lower misclassification score, although this comes at the cost of a larger percentage of samples assigned to multiple species. We have consciously decided for a more conservative classifier and we intend to resolve the samples which cannot be assigned to a single species on an ad hoc basis; either by more in depth analysis of specific amplicons (to be developed), by considering carefully documented species ranges where available (e.g. to distinguish between *An. bwambae* and *An. fontenillei*, both of which have very specific geographical areas where they occur) or by performing whole genome shotgun sequencing if the identity of the samples is of greater interest.

The UMAP projections can probably be tuned better to achieve similar assignment performance to the VAE, but UMAP does not outperform the VAE.

## Appendix 3

### ADMIXTURE in the *An. gambiae* complex

To assess the amount of population differentiation between the different species in the *An. gambiae* complex, we ran ADMIXTURE on the reference dataset GCref v1. We ran ADMIXTURE on the *8*-mer count tables; first we filtered for variable *8*-mers which take count values in {0,1,2} for all samples; there are 25,301 such *8*-mers. We treated these *8*-mers essentially as unlinked SNPs. We are aware that these input data do not exactly follow the assumptions behind the probabilistic model used by ADMIXTURE, but using only unlinked SNPs would allow for at most one SNP per amplicon, so 62 in total and it would be quite challenging to select one without bias to a certain species or group of species. Despite not following the model assumptions, we believe that ADMIXTURE used on k-mers should allow us to judge whether ADMIXTURE constitutes a useful approach to differentiate between species in the *An. gambiae* complex.

The inferred ADMIXTURE proportions for different values of K can be found in *Appendix 3— figure 1*. At K=3 and K=4, ADMIXTURE does a good job at distinguishing the well-represented species: *An. gambiae, An. coluzzii* and *An. arabiensis*. However, it struggles to differentiate between the species represented by a smaller number of samples, even though many of these species are much more diverged from each other than *An. gambiae* and *An. coluzzii* are (*Fontaine et al., 2015*).

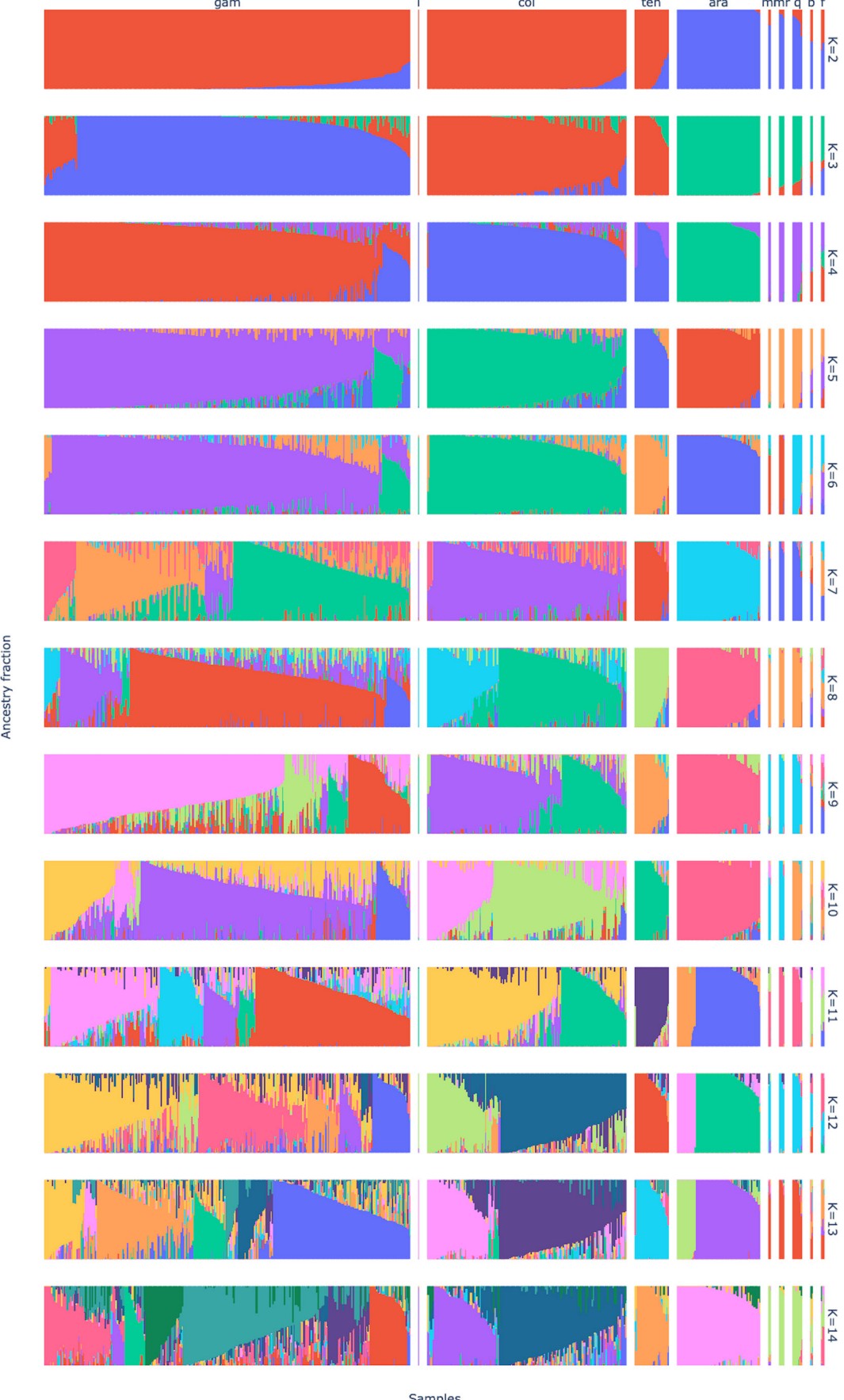

**Appendix 3—figure 1.** Admixture fractions of the GCref v1 samples for K from 2 to 14. The samples are ordered by species, from left to right *An. gambiae*, Agamp4 reference genome, *An. coluzzii*, *An. tengrela*, *An. arabiensis*, *An. melas*, *An. merus*, *An. quadriannulatus*, *An. bwambae* and *An. fontenillei*. The samples within a species are ordered by ancestry fraction and the within species ordering is not the same between rows.

For the higher values of K, ADMIXTURE reveals substructure and mixed ancestry in the *An. gambiae* and *An. coluzzii* samples. The first glimpse of substructure we see already at K=3, where the *An. gambiae* samples from Guinea-Bissau and the Gambia are modelled as a mixture of *An. gambiae* and *An. coluzzii*. This substructure is also found in the VAE projection.

While the *An. arabiensis* from Madagascar split out as a separate cluster in the VAE and in the three UMAP projections discussed in the next section, ADMIXTURE only identifies them as a separate group at K=11. So similarly as for the species represented by few samples, it seems like ADMIXTURE is hesitant to assign a unique ancestry to a small group of samples.

ADMIXTURE identifies some structure in the *An. gambiae* complex when treating variable *8*-mers as SNPs. It is particularly good at differentiating between groups represented by a larger number of samples and additionally finds some substructure within these groups. It is pronouncedly worse than the VAE at differentiating between species for which we have fewer samples. This is a shortcoming given that a major aim of applying ANOSPP widely to samples from across the globe is to identify novel species that may or may not be contributing to transmission.

To assess which values of K are most informative, we recorded the cross-validation error for these runs, see *Appendix 3—figure 2*. There are 9 species represented in the dataset GCref v1, so ideally we would find a minimal cross-validation error around K=9. The cross validation plot indicates that we need a minimum of about 9 populations, but it does not unambiguously indicate a suitable value of K. To properly assess the cross validation error, we'd have to run multiple replicates.

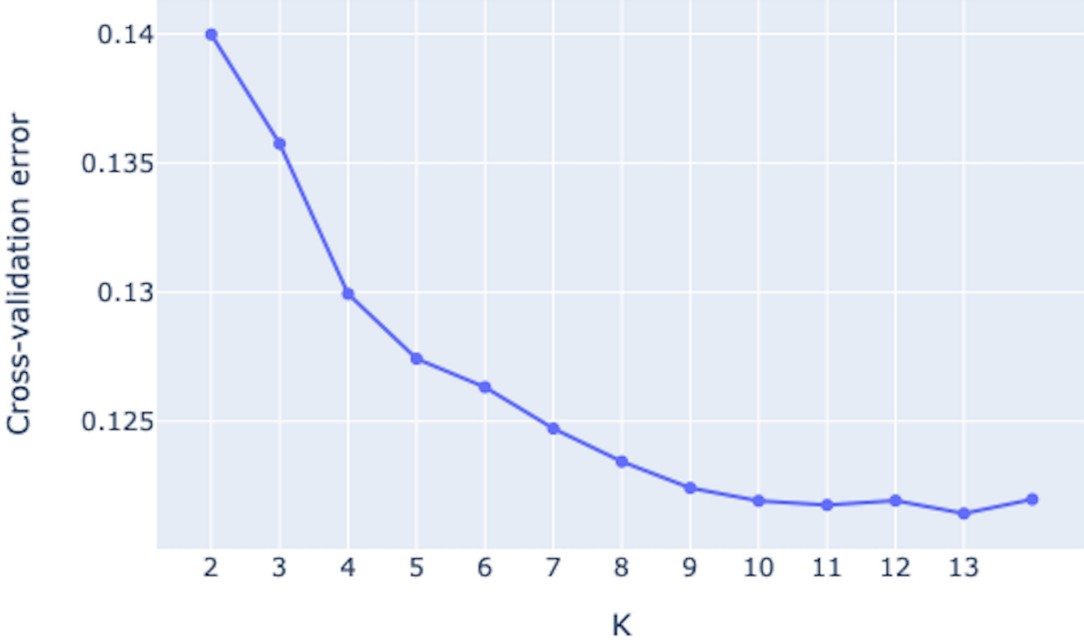

**Appendix 3—figure 2.** Cross-validation error for different values of K. Only one replicate shown.

Although ADMIXTURE does find structure in the *An. gambiae* complex, it is principally a method for assessing population differentiation, not for species assignment. It is possible to build an assignment method around it, but given that it performs worse than the VAE at differentiating between all species in the *An. gambiae* complex, we prefer using the VAE based method for assignments.

## Appendix 4

## Parameter choices for the VAE

Our VAE architecture is inspired by popVAE (*Battey et al., 2021*). We used their default parameter choices for learning rate, training iterations and validation proportion as well as the 'elu' propagation between the layers of the neural network and the 'linear' activation function to transform the output from the encoder to the latent space variables. Because we model $k$-mer counts as independent Poisson variables, we use the 'softplus' activation function to transform the output from the last layer of the decoder.

To set the width and depth of the encoder and decoder, we ran a grid search over a combination of widths and depths. We kept the width and depth the same for the decoder and the encoder and we trained the VAE with three different seeds for each width-depth combination. For high width values (more than 500 nodes) the run time required to train the VAE was very long and the process would often crash when conducted on a desktop. For very small networks (width 32, depth 4) the resolution was poor, but in the middle range the visible structure was not severely affected by the choice of width and depth, so we went with width 128 and depth 6, which gave a good resolution and reasonable run time.

As mentioned in the main text, the loss function contains a parameter, $w$, that controls the relative importance of the data driven term and the regularisation term. We ran a grid search ranging over four orders of magnitude, training with three different seeds for each search point. For small values of $w$, which gives higher weight to regularisation term, the resulting latent space dimension shows a strong correlation between the latent space dimensions, suggesting that only highly differentiated samples can overpower the effect from the regularisation term. For high values of $w$, the training process of the VAE becomes more unstable: it crashes sometimes and the resulting latent space projections look less similar to those obtained with the same $w$ value, but a different seed. We chose $w$ equal to 1000, which resulted in reasonably stable projections, with good visible structure.

To assess the effect of using $8$-mers over $k$-mers of other length, we ran the VAE with $6$-mers and with $10$-mers as input, see *Appendix 4—figure 1*. The projections using $6$-mers, in both two and three dimensions, show less tight and more overlapping clusters by species compared to the $8$-mer projection. There are only 4,096 unique $6$-mers, compared to >65,000 unique $8$-mers, and with almost 10 kB of sequence, there will be much fewer unique $6$-mers than unique $8$-mers. Although the VAE loss function is based on counts rather than presence/absence of $k$-mers, it probably is more sensitive to the difference between 0 and 1 than to the difference between 2 and 3. The VAE based on $10$-mers shows clusters quite similar to those found in the $8$-mer projection, although *An. quadriannulatus* is much closer to *An. bwambae* and *An. fontenillei*. It took much longer to train the $10$-mer projection (5.5 hr compared to 20 min for the $8$-mers), which is not a major drawback, because the training has to happen only once, but it does make it harder to tweak the parameters for the $10$-mer projection.

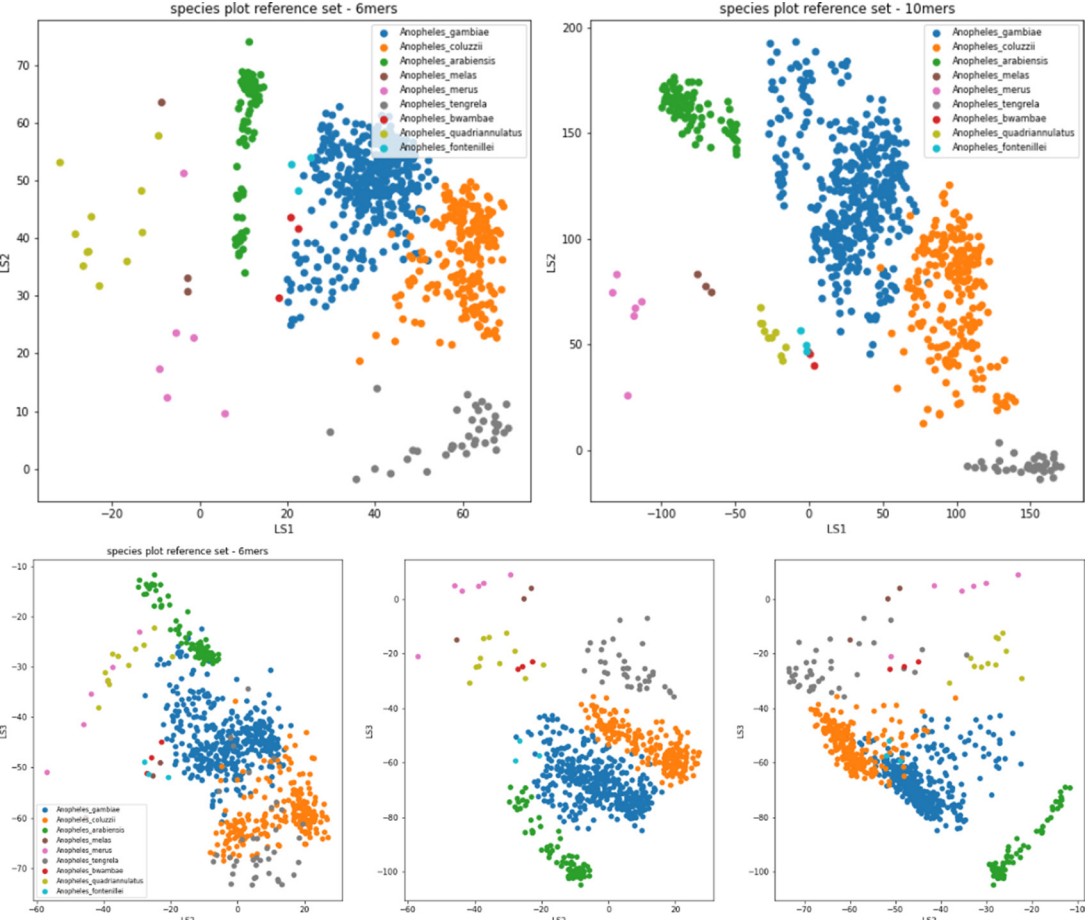

**Appendix 4—figure 1.** VAE projection of GCref v1 in two dimensions (top) using *6*-mers (left) and *10*-mers (right) and in three dimensions using *6*-mers (bottom).

To determine the number of latent space dimensions, we ran a grid search over 2, 3 and 4 latent space dimensions. In fact, the grid search for *w* and the number of latent space dimensions was run simultaneously, but there seemed to be little interaction between *w* and the number of latent space dimensions. For the training set containing only *An. gambiae*, *An. coluzzii* and *An. arabiensis*, adding a third or fourth dimension did not add to the visible structure, so in this case we opted for two latent space dimensions for visualisation purposes. However, for the full training set GCref v1, in the two-dimensional projection, several species clusters are very close to each other, making this hard to use for species assignment. Adding a third dimension resolves this problem, even though the second and third dimensions are strongly correlated (see *Figure 4*). The benefit of the strong correlation is that we can visualise the structure in two dimensions.

## Appendix 5

## Geographic stability

Here we test the robustness of our VAE projection by removing all samples collected in one geographic location from the training set and then projecting those samples and the validation set and comparing these results to those obtained using the VAE trained on the full training set. In particular, we will pay attention to the separation of the species clusters, the classification accuracy of the validation set and the visible structure within the species clusters.

For this analysis, we use a subset of the reference dataset GCref v1 and the validation set GCval v1, containing only the species *An. coluzzii*, *An. gambiae* and *An. arabiensis*, because for the other species in the *An. gambiae* complex we have very few samples, the samples in this dataset are collected at only one or two locations and the species have more limited geographic ranges than *An. coluzzii*, *An. gambiae* and *An. arabiensis*. With only these species, two latent space dimensions are sufficient to exhibit the relevant structure, so that is what we will use here. We have performed these drop-outs for a subset of the geographic locations, selecting for those we thought would be most likely to affect the structure of the projection.

Below, we discuss the drop-out experiments in more detail, but to summarise: overall, the separation of the species clusters remained intact and the ability to classify species of the samples in the training set and in the validation set was not significantly affected. Samples from The Gambia and Guinea-Bissau could be less accurately classified when no samples from this geographic region were included in training the VAE, but for all other locations we tested, the ability to classify species remained the same. The degree to which geographic structure is visible within the species clusters was somewhat affected for certain collection countries. However, the visible structure never completely disappeared, nor did we see a considerably different geographic structure, it was merely that the structure became a bit more blurred or more condensed than in the original projection.

## Angola

We removed 10 *An. coluzzii* samples from the training set, see *Appendix 5—figure 1*. The separability of species clusters is similar compared to the projection of the VAE trained on the full dataset. The geographic structure within the *An. gambiae* and *An. arabiensis* clusters remains intact, but the *An. coluzzii* cluster loses some of its visible structure. Angola is on the edge of the geographical range of the *An. coluzzii* samples in this dataset and these samples are also on the very top of the VAE projection, suggesting that they are responsible for a considerable amount of the structure within *An. coluzzii*.

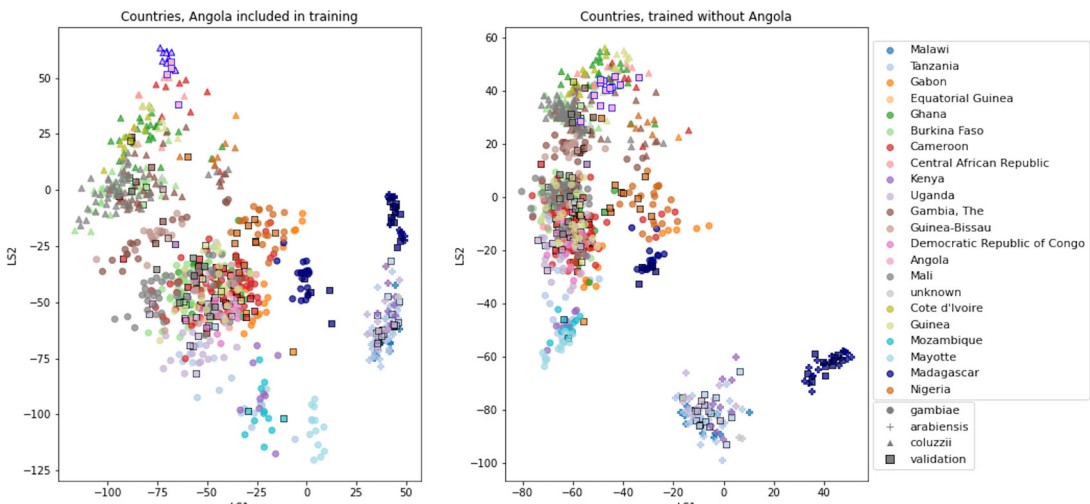

**Appendix 5—figure 1.** VAE projections of *An. arabiensis*, *An. coluzzii* and *An. gambiae* from different geographic locations. **Left:** samples from Angola included in VAE training. **Right:** samples from Angola excluded from VAE training. Samples are coloured by country of collection. Squares are validation samples (not used in VAE training), triangles are *An. coluzzii* individuals, circles are *An. gambiae* individuals and crosses are *An. arabiensis* individuals. Samples from Angola are highlighted with a blue edge, all other validation samples have a black edge.

## Cameroon

We removed 10 *An. coluzzii* and 66 *An. gambiae* from the training set, see *Appendix 5—figure 2*. The separability of the species clusters is similar compared to the projection of the VAE trained on the full dataset. The geographic structure within the *An. coluzzii* and *An. arabiensis* clusters remains intact, but the *An. gambiae* cluster loses some visible structure in its main subcluster, while the structure in the other four subclusters (formed by samples from The Gambia and Guinea-Bissau; by samples from Nigeria and Gabon; by samples from Madagascar and by samples from various East African countries respectively) remains largely intact. This is probably due to the fact that the samples from Cameroon form a large proportion of the samples in the main *An. gambiae* cluster.

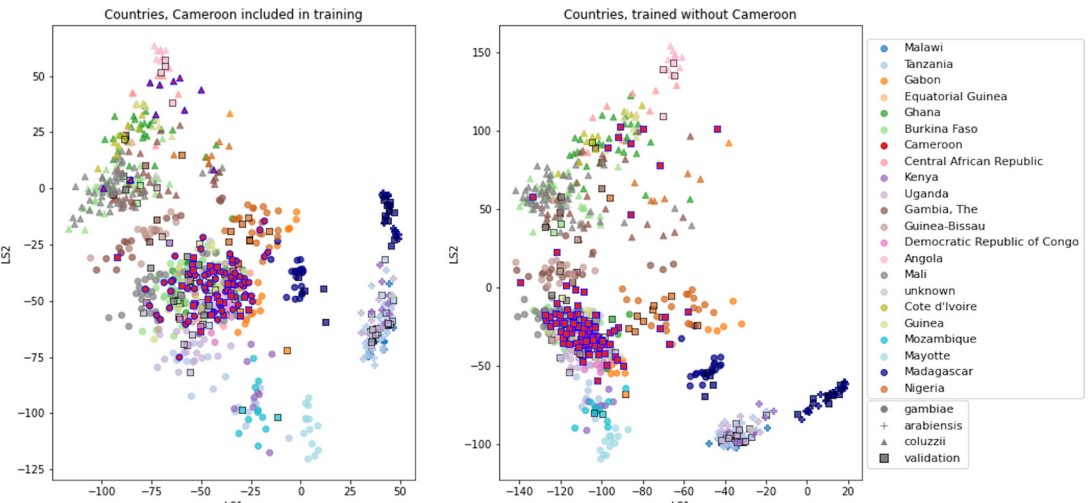

**Appendix 5—figure 2.** VAE projections of *An. arabiensis*, *An. coluzzii* and *An. gambiae* from different geographic locations. **Left:** samples from Cameroon included in VAE training. **Right:** samples from Cameroon excluded from VAE training. Samples are coloured by country of collection. Squares are validation samples (not used in VAE training), triangles are *An. coluzzii* individuals, circles are *An. gambiae* individuals and crosses are *An. arabiensis* individuals. Samples from Cameroon are highlighted with a blue edge, all other validation samples have a black edge.

## The Gambia and Guinea-Bissau

We removed 33 *An. coluzzii* and 19 *An. gambiae* from The Gambia and 24 *An. gambia* from Guinea-Bissau from the training set, see *Appendix 5—figure 3*. Considering the training samples, the separability of the species clusters is very clear. However, the classification accuracy of the projected samples from The Gambia and Guinea-Bissau is only 70%. This is probably due to the fact that these samples form the border between the *An. coluzzii* and *An. gambiae* clusters in the projection of the VAE trained on the complete dataset. So by leaving those samples out, we miss the important label information necessary to classify the edge cases. The subclusters of the *An. gambiae* cluster are arguably more pronounced than in the original projection, but the structure within the subclusters of the *An. gambiae* cluster and in the entire *An. coluzii* cluster reduces.

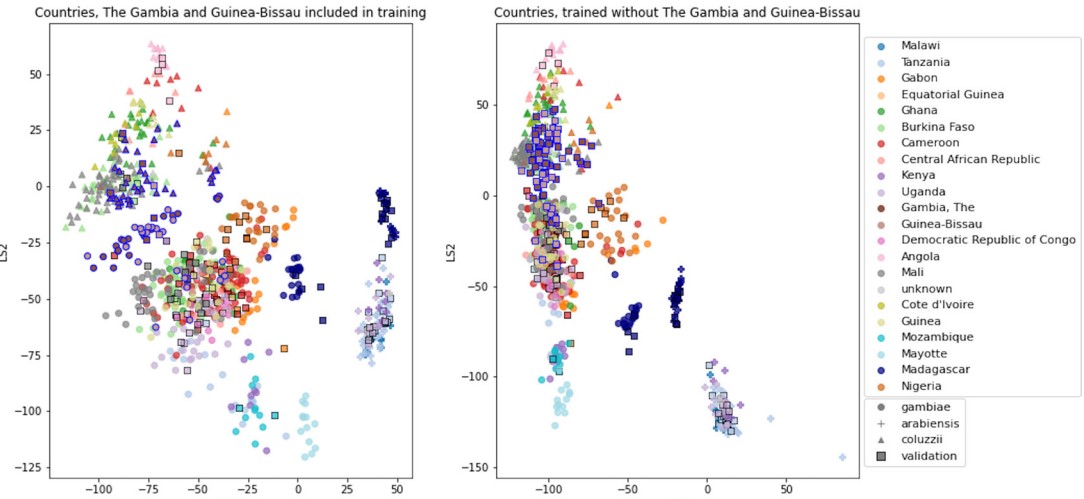

**Appendix 5—figure 3.** VAE projections of *An. arabiensis*, *An. coluzzii* and *An. gambiae* from different geographic locations. **Left:** samples from The Gambia and Guinea-Bissau included in VAE training. **Right:** samples from The Gambia and Guinea-Bissau excluded from VAE training. Samples are coloured by country of collection. Squares are validation samples (not used in VAE training), triangles are *An. coluzzii* individuals, circles are *An. gambiae* individuals and crosses are *An. arabiensis* individuals. Samples from The Gambia and Guinea-Bissau are highlighted with a blue edge, all other validation samples have a black edge.

## Madagascar

We removed 20 *An. arabiensis* and 18 *An. gambiae* from the training set, see *Appendix 5—figure 4*. The separability of the species clusters is similar compared to the projection of the VAE trained on the full dataset and the geographic structure within the three species clusters remains largely intact. When the Madagascar samples are included in training the VAE, they form a tight subcluster, both for *An. gambiae* and *An. arabiensis*. When they are not included in training, these clusters stand out less, due to the fact that the VAE does not recognise the features that make them stand out.

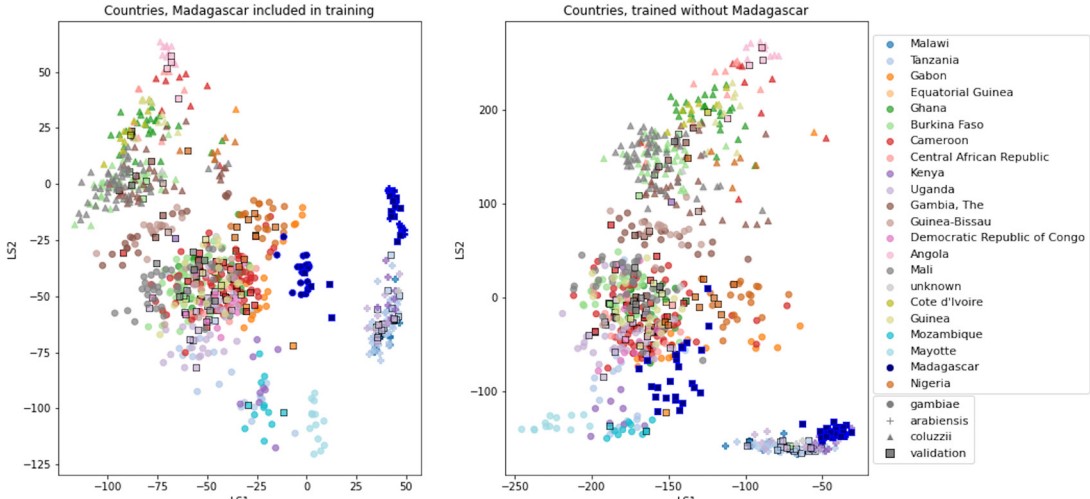

**Appendix 5—figure 4.** VAE projections of *An. arabiensis*, *An. coluzzii* and *An. gambiae* from different geographic locations. **Left:** samples from Madagascar included in VAE training. **Right:** samples from Madagascar excluded from VAE training. Samples are coloured by country of collection. Squares are validation samples (not used in VAE training), triangles are *An. coluzzii* individuals, circles are *An. gambiae* individuals and crosses are *An. arabiensis* individuals. Samples from Madagascar are highlighted with a blue edge, all other validation samples have a black edge.

## Mali

We removed 57 *An. coluzzii* and 43 *An. gambiae* from the training set, see *Appendix 5—figure 5*. The separability of the species clusters is similar compared to the projection of the VAE trained on the full dataset. The visible geographic structure within the three species clusters reduces a bit. This could be because we removed such a large number of samples.

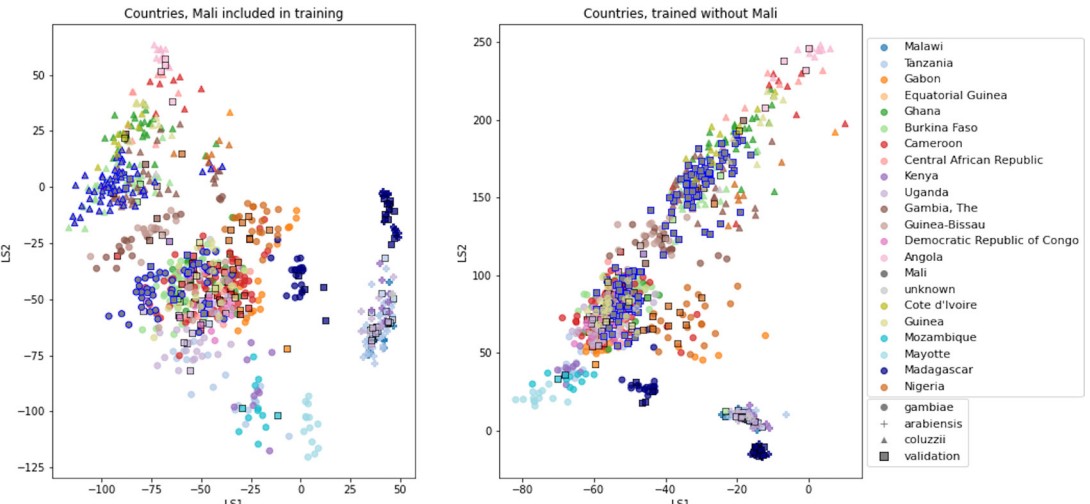

**Appendix 5—figure 5.** VAE projections of *An. arabiensis*, *An. coluzzii* and *An. gambiae* from different geographic locations. **Left:** samples from Mali included in VAE training. **Right:** samples from Mali excluded from VAE training. Samples are coloured by country of collection. Squares are validation samples (not used in VAE training), triangles are *An. coluzzii* individuals, circles are *An. gambiae* individuals and crosses are *An. arabiensis* individuals. Samples from Mali are highlighted with a blue edge, all other validation samples have a black edge.

## Nigeria

We removed 6 *An. coluzzii* and 20 *An. gambiae* from the training set, see *Appendix 5—figure 6*. The separability of the species clusters is similar compared to the projection of the VAE trained on the full dataset and the geographic structure within the three species clusters remains largely intact. When the Nigerian samples are included in training the VAE, the *An. gambiae* individuals are more separated from the main *An. gambiae* cluster than when they are projected using the VAE excluding them from training. This is because the VAE has not been trained to recognise the features that distinguish these Nigerian *An. gambiae* individuals from the other samples.

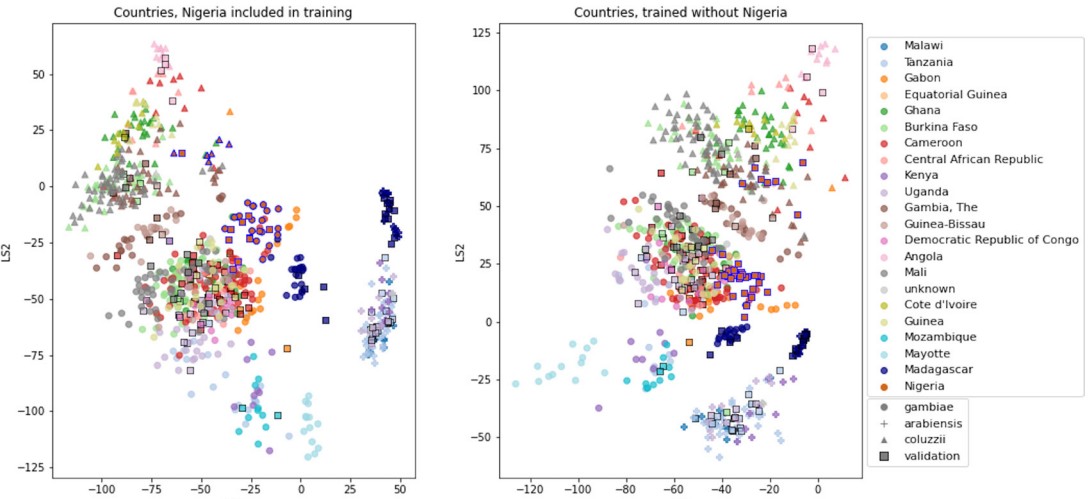

**Appendix 5—figure 6.** VAE projections of *An. arabiensis*, *An. coluzzii* and *An. gambiae* from different geographic locations. **Left:** samples from Nigeria included in VAE training. **Right:** samples from Nigeria excluded from VAE training. Samples are coloured by country of collection. Squares are validation samples (not used in VAE training), triangles are *An. coluzzii* individuals, circles are *An. gambiae* individuals and crosses are *An. arabiensis* individuals. Samples from Nigeria are highlighted with a blue edge, all other validation samples have a black edge.

## Tanzania

We removed 37 *An. arabiensis* and 20 *An. gambiae* from the training set, see *Appendix 5—figure 7*. The separability of the species clusters is similar compared to the projection of the VAE trained on the full dataset. The subclusters of *An. gambiae* and *An. arabiensis* remain well separated, but the visible geographic structure within the (sub)clusters reduces a bit.

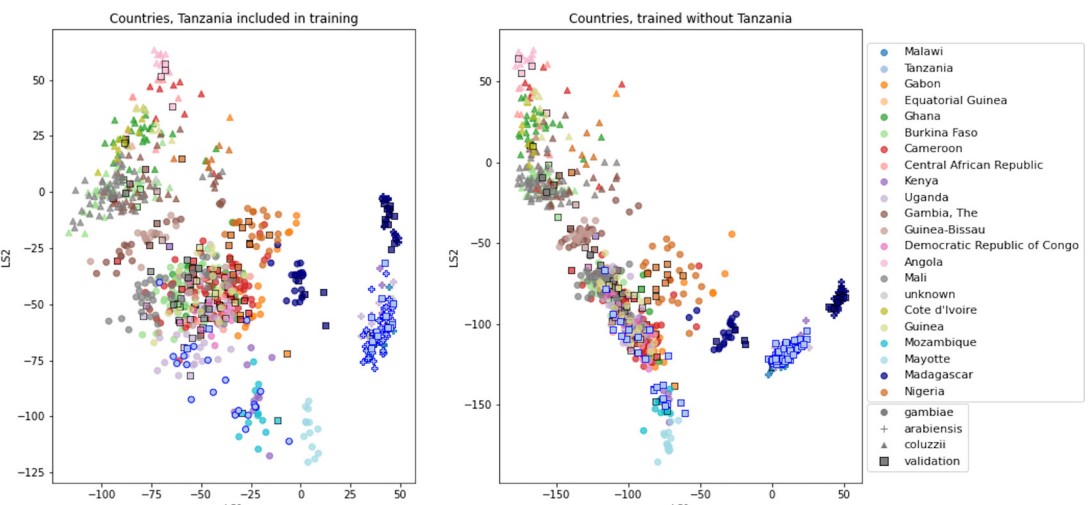

**Appendix 5—figure 7.** VAE projections of *An. arabiensis*, *An. coluzzii* and *An. gambiae* from different geographic locations. **Left:** samples from Tanzania included in VAE training. **Right:** samples from Tanzania excluded from VAE training. Samples are coloured by country of collection. Squares are validation samples (not used in VAE training), triangles are *An. coluzzii* individuals, circles are *An. gambiae* individuals and crosses are *An. arabiensis* individuals. Samples from Tanzania are highlighted with a blue edge, all other validation samples have a black edge.

