## [Editor Report]

This piece presents a new method, ANOSPP, that uses cheap sequencing to infer genealogical relationships between Anopheles individuals, including the possibility of species-level identification. The method is versatile and will be useful to vector biology researchers. The proof-of-principle presented in the manuscript is convincing and will serve as a blueprint for future studies in Anopheles.

---

## [Decision Letter]

**Decision letter after peer review:**

Thank you for submitting your article "High resolution species assignment of *Anopheles* mosquitoes using *k*-mer distances on targeted sequences" for consideration by *eLife*. Your article has been reviewed by 3 peer reviewers, one of whom is a member of our Board of Reviewing Editors, and the evaluation has been overseen by Dominique Soldati-Favre as the Senior Editor. The reviewers have opted to remain anonymous.

The reviewers have discussed their reviews with one another, and the Reviewing Editor has drafted this to help you prepare a revised submission. In general all reviewers saw merit in the submission but had some concerns regarding the scope of the work and the interest the findings can elicit in general audience not specialized in vectors. You will see that the reviewers had several comments that can make the manuscript a stronger piece. Collectively, all reviewers agreed on two improvements that would make the manuscript potentially suitable for *eLife*. I am looking forward to a resubmission of the piece.

Essential revisions:

1) All reviewers agree that the findings from this piece has the potential to benefit the vector community. The common comment from all reviewers was the need to more carefully delineate the scope and impact of the work. One way to do this is to be more straightforward on what instances this method can and should be used. This is of particular importance for the *eLife* audience which extends beyond anopheline researchers.

2) The main claim of the manuscript is that the method the authors are presenting is a major advance in species compared to the tools available in the field. This needs to be either further demonstrated by providing a comparison to other sequencing methods (MSG, RAD-Seq, etc), and computational tools (Structure, ADMIXTURE), or by tempering the claims.

*Reviewer #2 (Recommendations for the authors):*

First: this manuscript will be of interest to entomologists and vector biologists who do not have a background in machine learning. With that in mind, I commend the authors for their clear efforts in making the algorithmic and technical details of this manuscript understandable to a lay audience.

Three suggestions along those lines follow. First, the idea of a kmer table may not be immediately intuitive to someone who is not already familiar with them. I suggest making explicit (or more prominent, if I have missed this in the text) that the kmer table contains ALL possible kmers of that length. This is readily apparent in retrospect, but was not immediately clear.

Second, again, the authors have done a good job making what little math is present in this paper accessible to a wide audience, without resorting to jargon. However, considering this audience, it would be good to have the specific formal meaning of the vertical bars in both equations defined for those who are not familiar with set notation. At the first equation seems like a natural place for it.

Lastly on this topic, the kmer table contains the occurrence counts, and kmer distance is calculated based on the number of non-matching kmers. Is a matching kmer defined by shared presence, or an identical number of occurrences? Ie, if 8mer CCTGAAAT occurs twice in q1 and four times in q2, is this counted as a match for this entry in the table?

In addition, after reading the manuscript, I was curious about the potential effect of chromosomal inversions in the gambiae complex. The authors mention this as a type of variation that should be well-represented in the reference database. If the reference and validation sets naturally contain a reasonably polymorphic assortment of the inversions, it would appear that the method is robust to these, but this may be a strong assumption. Do the authors have any predictions about the potential consequences of inversions that could help researchers applying this method, or interpreting the resulting data, be wary of "edge cases?" Do a substantial proportion of amplicons overlap a common inversion in this complex?

Finally, in the discussion of An. nili and An. hyrcanus, as well as other relabeled specimens discussed in the supplement, the authors show how this tool can also generate phylogenetic questions. This is very interesting! However, given the known history of introgression in the gambiae and funestus complexes, and the probable history of such introgression elsewhere, it is also worth very briefly touching on the potential for this method to suggest patterns of relatedness between species that do not reflect the true phylogeny. This simply follows from interrogating markers genome-wide, and is not a drawback or criticism of the method; however, the clarification may be worthwhile for the sake of those interested in applying this method or re-using the resulting data.

*Reviewer #3 (Recommendations for the authors):*

My main concerns come from not being entirely certain what the proper scope is for such an investigation – I will elaborate a bit below, and I am totally happy to defer to the editor or other reviewers on what they feel is best here. But as a reader, I was particularly interested to get more detail on the advantages and disadvantages of different approaches (since the amplicon panel itself is already published elsewhere). Currently, it reads as a bit path dependent – the authors develop their nearest-neighbor scheme, show that it performs well at coarser scales but not finer scales, and then develop the VAE approach to assign within coarser groups.

In particular, I think it would be quite valuable to see what factors determine the best approach here. My understanding is that the regularization approach used by VAE leads to more continuous distributions in a space that tends towards the origin. For species assignment, this could be a good or a bad thing, I suppose – it might perform better for ambiguous samples but worse for well defined but subtly differentiated lineages? UMAP, in contrast, would be expected to yield more compact and spaced apart clusters, I believe – this might have different advantages and disadvantages. In Makunin 2022 UMAP doesn't seem to work particularly well – is the improvement here from the kmer approach or the VAE? It would be helpful to know what are the key advances that make things work so much better here.

I am sensitive to the desire to develop a practical tool here, so I am happy to hear if others disagree with this assessment. But if the goal is to more generally translate recent advances in clustering high-dimensional data to the species assignment context, I think a slightly broader scope in the presentation of results would help substantially. These recommendations are premised on potentially providing a broadened comparison of species assignment approaches for this specific context (i.e. large numbers of moderately divergent amplicons) – just suggestions of what I think would more helpfully contextualize this work, not preconditions for my recommendation to accept this manuscript.

I would be very curious to see something like a summary at the end with a more direct comparison of assignment accuracy with:

1. Read alignment with traditional nearest neighbor assignment – doesn't perform perfectly.

2. kmer tables with nearest neighbor performs incrementally better.

2a. Are 8mer tables actually the best tradeoff between robustness and sensitivity?

3. Machine learning approaches can improve on this.

3a. VAE

3b. UMAP or t-SNE?

This could provide more general guidelines that could be translated into other systems, increasing the utility of the work for the community.

---

## [Author Response]

Essential revisions:1) All reviewers agree that the findings from this piece has the potential to benefit the vector community. The common comment from all reviewers was the need to more carefully delineate the scope and impact of the work. One way to do this is to be more straightforward on what instances this method can and should be used. This is of particular importance for the eLife audience which extends beyond anopheline researchers.

We have more elaborately described potential use cases and why they are important in the conclusion.

2) The main claim of the manuscript is that the method the authors are presenting is a major advance in species compared to the tools available in the field. This needs to be either further demonstrated by providing a comparison to other sequencing methods (MSG, RAD-Seq, etc), and computational tools (Structure, ADMIXTURE), or by tempering the claims.

ANOSPP is designed to make species identification more accurate, more uniform across complexes, series and subgenera, while also being cheap and high throughput. So the panel really is an alternative for the current process of species identification, which includes morphological identification down to species complexes, followed by molecular typing (single locus sequencing or PCR/gel based diagnostics) to identify the species within the complex. Because ANOSPP works on the entire genus, it removes the need for morphological identification, the accuracy of which is dependent on the level of expertise of the taxonomist and the expected species diversity, and how well this is recorded, at the collection location. Within species complexes, our method is more reliable and nuanced than single locus barcoding approaches, because the information from 62 amplicon targets, spread over the genome, is combined. For a more elaborate discussion of our motivation to work with ANOSPP data over other sequencing approaches, see the reply to reviewer #1.

We compared the population structure within the *Anopheles gambiae* complex identified by the VAE to that identified by ADMIXTURE. ADMIXTURE identifies some structure in the *An. gambiae* complex when treating variable *8*-mers as SNPs. It is particularly good at differentiating between groups represented by a larger number of samples and additionally finds some substructure within these groups. However, it is pronouncedly worse than the VAE at differentiating between species for which we have fewer samples. For more elaborate discussion, see reply to reviewer #1, page 20 in the manuscript and newly added Appendix 3.

Reviewer #2 (Recommendations for the authors):First: this manuscript will be of interest to entomologists and vector biologists who do not have a background in machine learning. With that in mind, I commend the authors for their clear efforts in making the algorithmic and technical details of this manuscript understandable to a lay audience.Three suggestions along those lines follow. First, the idea of a kmer table may not be immediately intuitive to someone who is not already familiar with them. I suggest making explicit (or more prominent, if I have missed this in the text) that the kmer table contains ALL possible kmers of that length. This is readily apparent in retrospect, but was not immediately clear.

A toy 2-mer example is now included in the section on *k*-mers on page 7.

Second, again, the authors have done a good job making what little math is present in this paper accessible to a wide audience, without resorting to jargon. However, considering this audience, it would be good to have the specific formal meaning of the vertical bars in both equations defined for those who are not familiar with set notation. At the first equation seems like a natural place for it.

The notation for the size of a set has now been defined at its first occurrence.

Lastly on this topic, the kmer table contains the occurrence counts, and kmer distance is calculated based on the number of non-matching kmers. Is a matching kmer defined by shared presence, or an identical number of occurrences? Ie, if 8mer CCTGAAAT occurs twice in q1 and four times in q2, is this counted as a match for this entry in the table?

The definition for *k*-mer distance we use relies on the *k-*mer counts, so for two sequences to match exactly, the counts for each unique *k-*mer have to be equal. We have now highlighted this in the toy example and explicitly mentioned that the distance relies on counts and not mere presence/absence in the section on k-mers.

In addition, after reading the manuscript, I was curious about the potential effect of chromosomal inversions in the gambiae complex. The authors mention this as a type of variation that should be well-represented in the reference database. If the reference and validation sets naturally contain a reasonably polymorphic assortment of the inversions, it would appear that the method is robust to these, but this may be a strong assumption. Do the authors have any predictions about the potential consequences of inversions that could help researchers applying this method, or interpreting the resulting data, be wary of "edge cases?" Do a substantial proportion of amplicons overlap a common inversion in this complex?

We have looked into the 2La inversion in *An. gambiae* (this inversion is fixed in *An. arabiensis*). This is a reasonably large inversion (~20 Mb in the ~220 Mb genome) and four amplicons are located in the genomic region spanned by or closely neighbouring the inversion. We generated 2La karyotype predictions for the samples with whole genome data available using compkaryo (Love et al. *G3*, 2019) and found that two of these four amplicons contain variation that is correlated with the 2La karyotype. We used this variation to karyotype all the samples in the reference dataset GCref v1 and found that the karyotypes based on the amplicon data were in good agreement with the compkaryo results when both were available.

To explore whether the VAE contains any structure driven by the 2La inversion, we have to control for geographic population structure, because the 2La inversion frequencies vary along a latitudinal cline (e.g. Cheng et al. *Genetics*, 2010). So we looked at *An. gambiae* mosquitoes from two collection locations in Cameroon. These mosquitoes were collected in the exact same location, in the same month and year. As is shown in Author response image 1, the VAE projection does show structure correlated with the 2La karyotype. Bear in mind that this VAE is simply trained on all amplicons, so we are confident that we can develop tools that are tailored to accurate type 2La inversions from amplicon data.

We have not included this work in this manuscript. We need to develop and refine approaches to use the ANOSPP data to study features like inversions and this is still a work in progress.

**Author response image 1. sa2fig1:** 

Finally, in the discussion of An. nili and An. hyrcanus, as well as other relabeled specimens discussed in the supplement, the authors show how this tool can also generate phylogenetic questions. This is very interesting! However, given the known history of introgression in the gambiae and funestus complexes, and the probable history of such introgression elsewhere, it is also worth very briefly touching on the potential for this method to suggest patterns of relatedness between species that do not reflect the true phylogeny. This simply follows from interrogating markers genome-wide, and is not a drawback or criticism of the method; however, the clarification may be worthwhile for the sake of those interested in applying this method or re-using the resulting data.

We agree with the reviewer that this method is not suitable to resolve phylogenetic questions. However, it can be used to flag samples that deviate from our expectations, e.g. samples which were morphologically identified as the same species but are more divergent than expected. To truly resolve phylogenetic questions, we need whole genome data and appropriate reference genomes.

Reviewer #3 (Recommendations for the authors):My main concerns come from not being entirely certain what the proper scope is for such an investigation – I will elaborate a bit below, and I am totally happy to defer to the editor or other reviewers on what they feel is best here. But as a reader, I was particularly interested to get more detail on the advantages and disadvantages of different approaches (since the amplicon panel itself is already published elsewhere). Currently, it reads as a bit path dependent – the authors develop their nearest-neighbor scheme, show that it performs well at coarser scales but not finer scales, and then develop the VAE approach to assign within coarser groups.

This is a valid critique. However, our priority was to develop a functional method to perform species assignment with this amplicon panel. For a structured study to obtain the optimal assignment method, we’d have to investigate which encodings, projections and clustering methods work best for these data. This would be very interesting, but quite a tour de force. Below we have addressed the suggestions raised in this review and we have conducted the experiments which we consider to be within the scope of this manuscript.

In particular, I think it would be quite valuable to see what factors determine the best approach here. My understanding is that the regularization approach used by VAE leads to more continuous distributions in a space that tends towards the origin. For species assignment, this could be a good or a bad thing, I suppose – it might perform better for ambiguous samples but worse for well defined but subtly differentiated lineages? UMAP, in contrast, would be expected to yield more compact and spaced apart clusters, I believe – this might have different advantages and disadvantages. In Makunin 2022 UMAP doesn't seem to work particularly well – is the improvement here from the kmer approach or the VAE? It would be helpful to know what are the key advances that make things work so much better here.

To address this question we have generated projections for the *An. gambiae* complex with UMAP. As for the VAE, we used 8-mer count tables as input. The UMAP projections do indeed show tighter clusters which are more spaced apart, compared to the clusters found in the VAE latent space. This removes the need for an assignment method that deals with the ‘fuzzy boundaries’ between clusters, like the convex hull assignment method, but it comes at the cost of some hard misclassifications (i.e. the UMAP cluster label is different from the label assigned from whole genome data) for which is is impossible to attach to samples a measure of uncertainty in cluster membership based on their position in the projection. See page 20 in the manuscript and the new Appendix 2 for the full description of the experiment and the comparison to the VAE assignments.

I am sensitive to the desire to develop a practical tool here, so I am happy to hear if others disagree with this assessment. But if the goal is to more generally translate recent advances in clustering high-dimensional data to the species assignment context, I think a slightly broader scope in the presentation of results would help substantially. These recommendations are premised on potentially providing a broadened comparison of species assignment approaches for this specific context (i.e. large numbers of moderately divergent amplicons) – just suggestions of what I think would more helpfully contextualize this work, not preconditions for my recommendation to accept this manuscript.I would be very curious to see something like a summary at the end with a more direct comparison of assignment accuracy with:1. Read alignment with traditional nearest neighbor assignment – doesn't perform perfectly.

We have not generated alignments for all of our test and training samples, because this presents a whole set of its own challenges and many of the amplicons will have extremely poor alignments across the wide set of species used here. If the alignment algorithm introduces big blocks of insertions and deletions to deal with highly diverged sequences, it is unclear what is a fair way to call variants. If instead of using multiple sequence alignments, we aligned to a reference genome, we’d have to choose which reference genome to use and how to deal with reference bias, which would be substantial – we’d likely have to use many different reference genomes. Because of these issues, and because the assignment performs better than the method presented in Makunin et al. 2022, we prefer to use *k-*mer based methods.

2. kmer tables with nearest neighbor performs incrementally better.2a. Are 8mer tables actually the best tradeoff between robustness and sensitivity?

Our motivation for using *8-*mers is outlined on page 6 of the manuscript. We have also added a comparison between *6*-mers, *8-*mers and *10*-mers as VAE input in Appendix 4.

In the manuscript we demonstrate that the nearest neighbour method on *k-*mers can confidently distinguish most sufficiently diverged species. However, it generally doesn’t meet the assignment threshold of 0.8 for the samples within a species complex; see figure 3. Machine learning methods can improve the species-level assignments, while also providing a measure of uncertainty; which is important given the possibility of hybridisation. However, most machine learning methods require a large dataset of labelled samples, which is not available for every species complex.

3. Machine learning approaches can improve on this.3a. VAE3b. UMAP or t-SNE?This could provide more general guidelines that could be translated into other systems, increasing the utility of the work for the community.

We compared assignments using VAE against assignments using UMAP (see above, and manuscript page 20 and Appendix 2). We did not compare against t-SNE projections, because t-SNE cannot learn a projection function from a set of training samples and then use that to map new samples onto the same projection. This introduces a dependency of the projection on the test set and makes it much harder to understand and quantify the reliability of the assignments. We added a section on this to the main text on page 19.